# Characterization of a membrane-bound C-glucosyltransferase responsible for carminic acid biosynthesis in *Dactylopius coccus* Costa

Rubini Kannangara[1,2], Lina Siukstaite[1], Jonas Borch-Jensen[3], Bjørn Madsen[2], Kenneth T. Kongstad [4], Dan Staerk [4], Mads Bennedsen[2], Finn T. Okkels[2,6], Silas A. Rasmussen[5], Thomas O. Larsen[5], Rasmus J.N. Frandsen[5] & Birger Lindberg Møller [1]

Carminic acid, a glucosylated anthraquinone found in scale insects like *Dactylopius coccus*, has since ancient times been used as a red colorant in various applications. Here we show that a membrane-bound C-glucosyltransferase, isolated from *D. coccus* and designated DcUGT2, catalyzes the glucosylation of flavokermesic acid and kermesic acid into their respective C-glucosides dcII and carminic acid. DcUGT2 is predicted to be a type I integral endoplasmic reticulum (ER) membrane protein, containing a cleavable N-terminal signal peptide and a C-terminal transmembrane helix that anchors the protein to the ER, followed by a short cytoplasmic tail. DcUGT2 is found to be heavily glycosylated. Truncated DcUGT2 proteins synthesized in yeast indicate the presence of an internal ER-targeting signal. The cleavable N-terminal signal peptide is shown to be essential for the activity of DcUGT2, whereas the transmembrane helix/cytoplasmic domains, although important, are not crucial for its catalytic function.

[1] Plant Biochemistry Laboratory, Department of Plant and Environmental Sciences, University of Copenhagen, Thorvaldsensvej 40, 1871 Frederiksberg C, Denmark. [2] Chr. Hansen A/S, Bøge Alle 10-12, 2970 Hørsholm, Denmark. [3] VILLUM Center For Bioanalytical Sciences, Department of Biochemistry and Molecular Biology, University of Southern Denmark, 5230 Odense M, Denmark. [4] Department of Drug Design and Pharmacology, University of Copenhagen, Universitetsparken 2, 2100 Copenhagen, Denmark. [5] Department of Biotechnology and Biomedicine, The Technical University of Denmark, Søltofts Plads Building 221 and 223, 2800 Kgs. Lyngby, Denmark. [6] Present address: ActaBio ApS, Kongemarken 11, 4000 Roskilde, Denmark. Correspondence and requests for materials should be addressed to R.K. (email: dkruka@chr-hansen.com) or to B.L.M. (email: blm@plen.ku.dk)

Carminic acid (CA) is a natural red pigment found in some scale insects including the American cochineals (*Dactylopius coccus* Costa), which are native to tropical and subtropical regions of South America and Mexico. These scale insects are mostly sessile phloem feeders, living as parasites exclusively on cacti belonging to the *Opuntia* genus. In adult females of *D. coccus*, CA makes up 14–26% of the dry weight, while for adult males the yield is insignificant[1,2]. CA is thought to serve as a defense compound in the scale insect and has been shown to act as a potent feeding deterrent to ants[3]. Some predators that feed on *D. coccus*, like the larvae of the coccinellid beetle, *Hyperaspis trifurcata* and the *Laetilia coccidivora* moth, are able to sequester CA for use in their own defense[3,4].

*D. coccus*, which was used as colorant by the Mayans, Incas, and Aztecs, has since ancient times been an economically important insect[5]. During the Spanish colonial era, the dried insect powder was exported to Europe where it became an important commodity. By the early nineteenth century, the American cochineal was introduced and reared on its cactus host outside Central and South America. Today Peru is the main producer of CA from *D. coccus*[6].

CA production is very labor-intensive and involves rearing the *D. coccus* insects in large cacti plantations prior to harvesting and drying the adult females. The pigment is then extracted by boiling the dried insects in an alkaline solution. After removal of insect parts, the cochineal extract may be used as colorant. CA is highly soluble in water and can be used within the broad pH range from 2 to 9, and depending on the acidity or basicity of the solution, colors from orange to red and violet may be obtained. CA may be further purified by precipitation with alum under acidic conditions to produce a more intense red aluminum salt called carmine lake. This carmine lake is essentially water and acid insoluble, although water-soluble forms have been generated for applications demanding lower pH[7]. The CA pigment is widely used as a colorant in the food, textile, cosmetic, paint, and coating industry. Some of its beneficial attributes for industrial application include its high stability to heat and light, resistance to oxidation, non-carcinogenicity, and non-toxicity to humans upon skin contact or ingestion[7,8].

With the greater awareness about the impact of foods on human health and wellness, consumers are demanding natural colors as opposed to those produced synthetically. Many of the red synthetic colors have been shown to have adverse health effects. In light of this, CA has gained renewed popularity as a safe colorant with superior stability.

The biosynthetic pathway of CA in *D. coccus* has remained elusive. Based on its molecular structure, CA is classified as an anthraquinone glucoside. Two biosynthetic pathway routes may be envisioned for the formation of anthraquinones[9]. One route involves formation of the anthraquinone via a polyketide-based pathway, whereas the other route entails a shikimate-based pathway. The CA pigment has been proposed to be derived from a polyketide-based pathway, although no experimental evidence for such a route has been demonstrated in *D. coccus*[10,11]. The proposed route of biosynthesis starts with a stepwise condensation of 1 acetate and 7 malonate units to generate a hypothetical octaketide in a process catalyzed by a putative polyketide synthase (PKS) (Fig. 1). The octaketide is then cyclized to a presumable unstable anthrone that may undergo enzymatic or spontaneous oxidation to form the anthraquinone, flavokermesic acid (FK). Hydroxylation of FK results in the formation of kermesic acid (KA) which upon C-glucosylation affords CA. Small amounts of flavokermesic acid-C-glucosides (dcII) are present in metabolite extracts of *D. coccus*, implying that C-glucosylation could occur at the level of FK[12–14].

In parallel with the proposed pathway discussed above, some of the enzymes catalyzing the synthesis of CA have been hypothesized to originate from a *D. coccus* endosymbiont[10]. This hypothesis is attractive because polyketides are known to be widely produced in microorganisms and because *D. coccus* does not appear to sequester the CA from its *Opuntia* food/host plant[9]. In contrast to the situation in bacteria, fungi, and plants, limited molecular information on genes and enzymes responsible for polyketide biosynthesis is available from insects. The studies so far reported address the polyketide, pederin, which is found in beetles of *Paederus* sp. and *Paederidus* sp. Pederin is produced by an endosymbiotic bacterium and not by the insects[15–19].

In the current study, we characterize the membrane-bound UDP-glucosyltransferase (UGT), DcUGT2, which is responsible for catalyzing C-glucosylation of FK and KA to produce dcII and CA, respectively. The experimental approach involves classical protein fractionation of a detergent-solubilized *D. coccus* membrane fraction guided by transcriptomic and proteomics data and heterologous expression of candidate genes in *Saccharomyces cerevisiae*. DcUGT2 is predicted to be an endoplasmic reticulum (ER)-bound protein with the N-terminal part facing the lumen of the ER. Prediction analyses indicate that the protein has a cleavable signal peptide in the N terminus, a single transmembrane helix in the C terminus, and three potential N-glycosylation sites. Activity studies of truncated forms of the DcUGT2 enzyme suggest that targeting of the protein to the ER is essential for its activity.

## Results

**Establishing a *D. coccus* transcriptome.** A *D. coccus* transcriptomic profile was generated to identify putative UGTs involved in CA biosynthesis. Copious amounts of CA are present in adult female cochineals and it was assumed that genes encoding enzymes involved in the biosynthesis of this red pigment would therefore be highly expressed at this life stage. An Illumina sequencing analysis with 100-fold coverage of the polyadenylated RNA isolated from adult female cochineals was performed to identify putative *UGT* transcripts belonging to glycosyltransferase family 1. A total of 100,823,364 reads were generated with an average length of 89 bp, of which 74,434,099 reads passed the initial quality control. The passed reads were de novo assembled resulting in 35,154 contigs, representing different splice forms, partial and full-length transcripts. Annotation based on Pfam and on protein homology BLAST analyses identified 31 putative UGT candidates, of which four were predicted to be full-length and the rest partial. The four full-length sequences (*DcUGT1*, *DcUGT2*, *DcUGT4*, and *DcUGT8*) were among the 21 highest expressed putative *UGT* transcripts in adult female *D. coccus* insects displaying RPKM (Reads per Kilobase sequence per Million mapped reads) values of 108, 182, 54, and 10, respectively (Supplementary Data 1). An attempt to express the four full-length native *DcUGT* cDNAs were carried out in *S. cerevisiae* and *Aspergillus nidulans* with and without a C-terminal Strep-tag II (Strep) epitope. Transformants were confirmed by PCR followed by DNA sequencing of the amplified product, but no functional UGT activity could be measured. In this set of experiments, the UGT activity was monitored in soluble and microsomal protein extracts from the transformed heterologous hosts and from untransformed host controls. No product formation was detected in the liquid chromatography-mass spectrometry (LC-MS) and thin-layer chromatography (TLC) profiles following incubation with UDP-glucose or [$^{14}$C]UDP-glucose, respectively, and using the putative substrates FK and KA. FK and KA were supplied in the form of an isolated metabolite fraction from *Kermes vermilio*, a scale insect species incapable of producing dcII and CA. The

lack of a UGT activity prompted us to test for heterologous protein production after induced expression of the epitope-tagged *DcUGT* versions. Western blot analysis of total proteins extracted

from cultured transformants did not uncover any immunoreactive proteins. Thus, the absence of heterologous UGT activity was ascribed to either non-optimal codon usage of the native

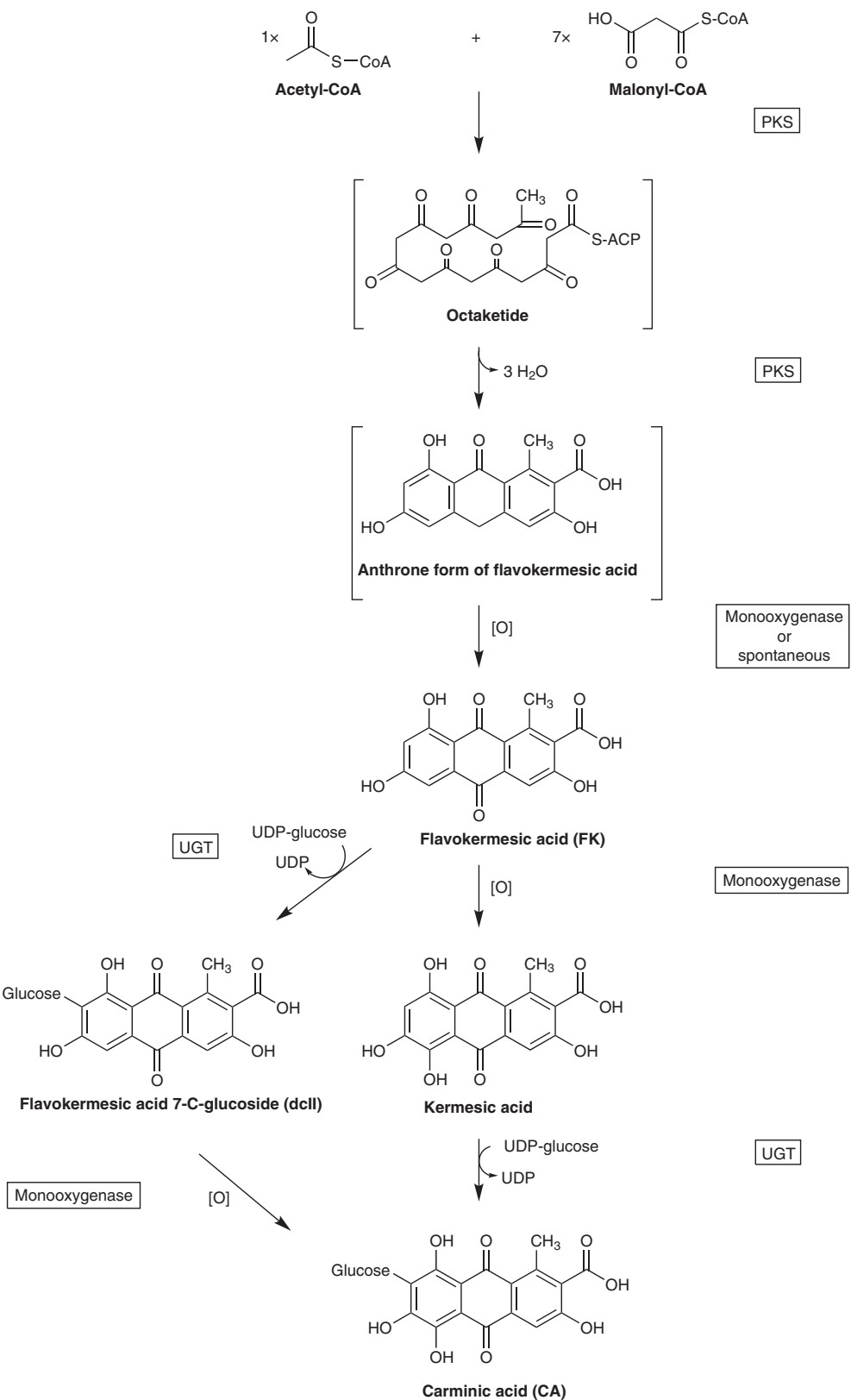

**Fig. 1** Putative carminic acid pathway in *Dactylopius coccus*. UDP uridine diphosphate, UGT uridine diphosphate glucosyltransferase, PKS polyketide synthase, CoA coenzyme A, [O] oxidizing agent, ACP acyl carrier protein

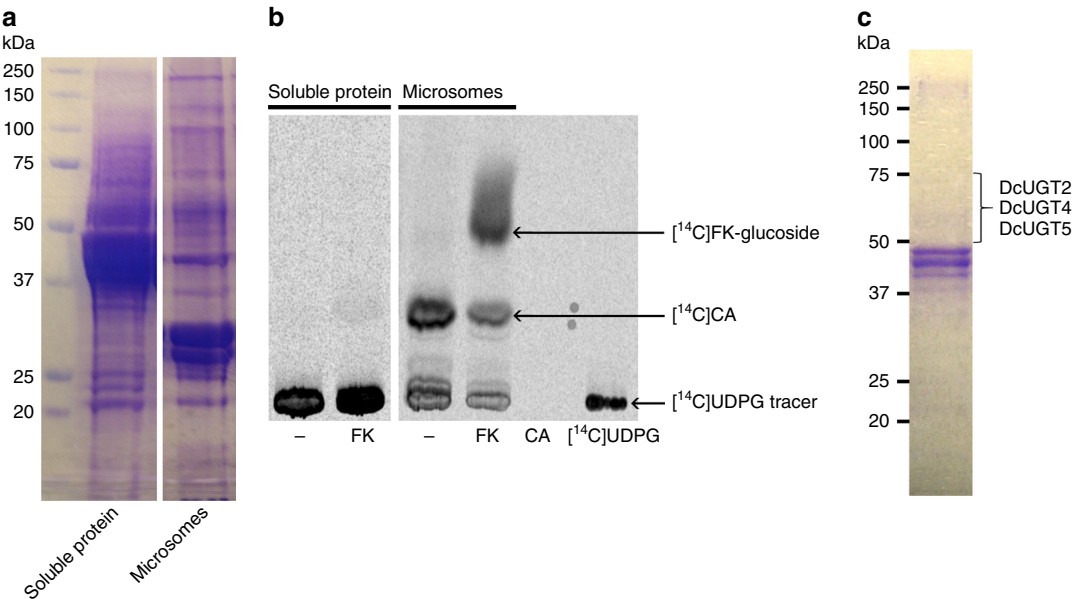

**Fig. 2** Identification of a *Dactylopius coccus* FK/KA-specific UGT activity. An isolated microsomal protein fraction and a soluble protein fraction from *D. coccus* were tested for glucosylation activity in vitro using the flavokermesic acid aglucone and the [$^{14}$C]UDP-glucose donor. **a** Coomassie-stained SDS gel of separated microsomal/soluble protein from *D. coccus*. **b** TLC-separated [$^{14}$C]-labeled products, formed in vitro and monitored by phosphorimaging. [$^{14}$C]UDPG [$^{14}$C]UDP-glucose, FK flavokermesic acid, CA carminic acid; − incubation without aglucone substrate. The in vitro formation of [$^{14}$C]CA was ascribed to the conversion of kermesic acid that still was bound to the *D. coccus* microsomes despite numerous wash steps during preparation. **c** A membrane-bound enzyme activity catalyzing the glucosylation of flavokermesic acid and kermesic acid was partially purified by anion-exchange chromatography after solubilization with reduced Triton X-100 (Supplementary Fig. 1). A fraction eluted with 100 mM NaCl and enriched with flavokermesic acid/kermesic acid-specific glucosylation activity was separated on an SDS gel followed by Coomassie staining. Proteins within the apparent mass region of 50–75 kDa were in-gel digested with trypsin and analyzed by LC-MS/MS. Tryptic peptides of DcUGT2, DcUGT4 and DcUGT5 were identified

*DcUGT* cDNA sequences, hampered transcription, or an instability/degradation of the foreign *DcUGT* transcripts. These negative results dictated initiation of a biochemical approach.

**Isolation of a *D. coccus* glucosylation activity**. A soluble and a microsomal protein fraction were isolated from fresh adult *D. coccus* females and tested for glucosylation activity towards FK with [$^{14}$C]UDP-glucose present as the sugar donor (Fig. 2a, b). An enzyme activity specifically capable of glucosylating FK and KA was present in the microsomal protein fraction (Fig. 2b). The observed formation of [$^{14}$C]CA in reactions supplemented with FK and in control reactions without added substrate is ascribed to the presence of carryover of KA bound to the microsomes in spite of introduction of several washing steps in the isolation procedure. Partial purification of the membrane-bound enzyme activity, responsible for glucosylating FK and KA, was accomplished following initial solubilization of the microsomal protein fraction using reduced Triton X-100 and then separation by anion-exchange chromatography using Q-Sepharose and application of a stepwise NaCl gradient (100–500 mM) (Supplementary Fig. 1). Fractions showing FK/KA-specific glucosylation activity were obtained following elution with 100 and 200 mM NaCl. UGT enzymes have masses within the range of 50–75 kDa. Based on the presence of the desired enzyme activity and sodium dodecyl sulfate-polyacrylamide gel electrophoresis (SDS-PAGE) analysis, protein fraction 1 was selected for further analysis. In comparison to other active protein fractions, it contained fewer proteins in the 50 to 75 kDa mass region (Supplementary Fig. 1). We expected that a reduced number of non-relevant proteins would optimize identification of the UGT responsible for the observed activity. Thus, protein fraction 1 was separated by SDS-PAGE and the proteins migrating in the 50 to 75 kDa region were subjected to

in-gel trypsin digestion, LC-MS/MS-based amino acid sequencing of the fragments obtained, and database searching (Fig. 2c). Tryptic peptide sequence hits, with a coverage of 45%, 8%, and 22% corresponding to DcUGT2, DcUGT4, and DcUGT5, respectively, were found when compared to the transcriptomic dataset and BLAST searches (Supplementary Data 2).

**Heterologous expression of *DcUGT* genes in yeast**. To determine whether any of the three UGTs found in the *D. coccus* membrane protein fraction were able to catalyze glucosylation of FK and KA in vitro, the three candidate genes were codon optimized and expressed with a Strep in *S. cerevisiae*. Prior to this, the full-length cDNA sequences were obtained for *DcUGT5* and confirmed for *DcUGT2* and *DcUGT4* from *D. coccus* by rapid amplification of cDNA ends (RACE). Five independent yeast transformants were selected for each *UGT* construct and microsomes were prepared from their cell cultures following galactose-induced protein expression. Western blot analysis using anti-Strep antibody detected the DcUGT2-Strep and DcUGT5-Strep proteins, but not the DcUGT4-Strep protein, indicating that induced synthesis of two of the three UGT candidates had been achieved in yeast (Fig. 3). Interestingly, the immunoreactive DcUGT2-Strep protein migrated with an apparent molecular mass of approximately 52 kDa which is smaller than its calculated mass of 58 kDa. In contrast, yeast microsomes containing DcUGT5-Strep gave rise to three distinct immunoreactive bands, of which one matched its calculated molecular mass of 59 kDa (Fig. 3). The two other immunoreactive polypeptides with lower masses were considered to be degradation products of the full-length DcUGT5-Strep protein. This profile was observed for all five transformants carrying the *DcUGT5-Strep* gene (Fig. 3).

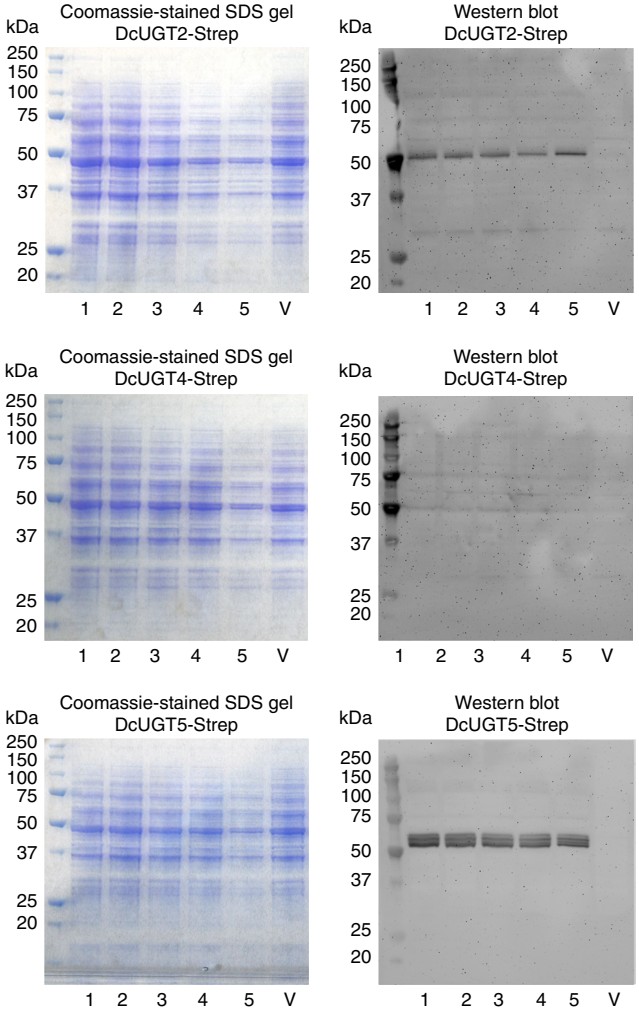

**Fig. 3** Heterologous expression *DcUGT* candidates in yeast. Five individual *S. cerevisiae* transformants were cultured for each Strep-tagged *UGT* candidate, *DcUGT2-Strep*, *DcUGT4-Strep* and *DcUGT5-Strep* along with a transformant carrying the *pYES-DEST52* vector. After galactose-induced heterologous protein expression, microsomal proteins were isolated from each transformant and separated on an SDS gel followed by either Coomassie staining or western blotting using an anti-Strep antibody. V pYES-DEST52 vector. Note that the anti-strep antibody reacts with the marker protein bands

Two selected yeast transformants, each carrying *DcUGT2-Strep* and *DcUGT5-Strep*, respectively, were analyzed for their in vitro glucosylation activity (Figs. 3 and 4b). FK was tested as the aglucone substrate using the structurally similar anthraquinone, asperthecin, as a control (Fig. 4a). This showed that only DcUGT2-Strep was able to catalyze production of radiolabeled glucosylated FK when incubated with [14C]UDP-glucose and the FK aglucone. The [14C] product corresponded to the compound formed with a *D. coccus* microsomal protein fraction after incubation with the same substrate, suggesting that DcUGT2 is the enzyme or at least one of those enzymes responsible for the FK-specific glucosylation activity observed in *D. coccus* (Fig. 4b). Assay products, generated in vitro with *D. coccus* microsomes, were treated with viscozyme to assess whether the [14C]FK-glucoside formed was caused by O- or C-glucosylation (Supplementary Fig. 2). In contrast to the O-glucoside control [14C] Linamarin, [14C]FK-glucoside was resistant to the viscozyme treatment and migrated with a similar relative migration (Rf) value as dcII, indicating that it was a C-glucoside. The *D. coccus*

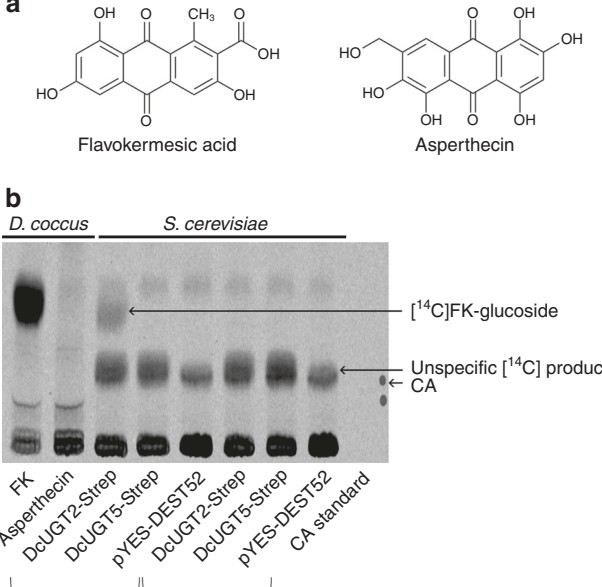

**Fig. 4** Identification of a DcUGT glucosylating FK in vitro. Microsomes of *D. coccus* and of yeast expressing Strep-tagged *DcUGT2*, *DcUGT5* and pYES-DEST52 vector, respectively, were incubated with [14C]UDP-glucose and with/without flavokermesic acid. As a control, *D. coccus* microsomes were incubated with the asperthecin aglucone and [14C]UDP-glucose. Formed products were separated by TLC. **a** Aglucone substrates tested in the in vitro glucosylation assay. **b** TLC-separated [14C] products formed in vitro and viewed by phosphorimaging. FK flavokermesic acid, CA carminic acid

microsomal proteins were incapable of glucosylating asperthecin in vitro, implying that the configuration of functional groups on the anthraquinone backbone was critical for C-glucosylation to occur. As observed from the vector control, *S. cerevisiae* also possesses endogenous glucosylation activities, yielding a [14C] product with a similar Rf value as CA in the applied TLC system (Fig. 4).

In order to distinguish CA from this unknown [14C] product and to obtain further confirmation of the structures of the glucosylated products formed, the enzyme-generated products were analyzed by high-performance liquid chromatography-MS (HPLC-MS) (Fig. 5). Yeast microsomes harboring DcUGT2-Strep were incubated with UDP-glucose in the presence of KA and FK as substrates. This corroborated the ability of DcUGT2-Strep to glucosylate FK and clearly demonstrated that KA acted as an acceptor molecule for DcUGT2-Strep when compared with the negative vector control (Fig. 5). The formed glucosides were identified as being the C-glucosides dcII and CA based on their MS fragmentation patterns and comparison with authentic standards (Fig. 5).

**Characterization of DcUGT2.** To characterize the kinetic properties of DcUGT2, yeast microsomal membranes containing heterologously produced DcUGT2 were solubilized using reduced Triton X-100 and DcUGT2 was affinity purified by its Strep-tag II (Supplementary Fig. 3 and Fig. 6). In vitro tests of the isolated Strep-tagged DcUGT2 showed that its enzymatic activity was lost upon isolation (Fig. 6). Thus, it was not possible to carry out a more detailed kinetic study on DcUGT2-Strep because of its labile nature.

The *DcUGT2* transcript from *D. coccus* encodes a 515-amino acid long protein that is predicted to be membrane bound as

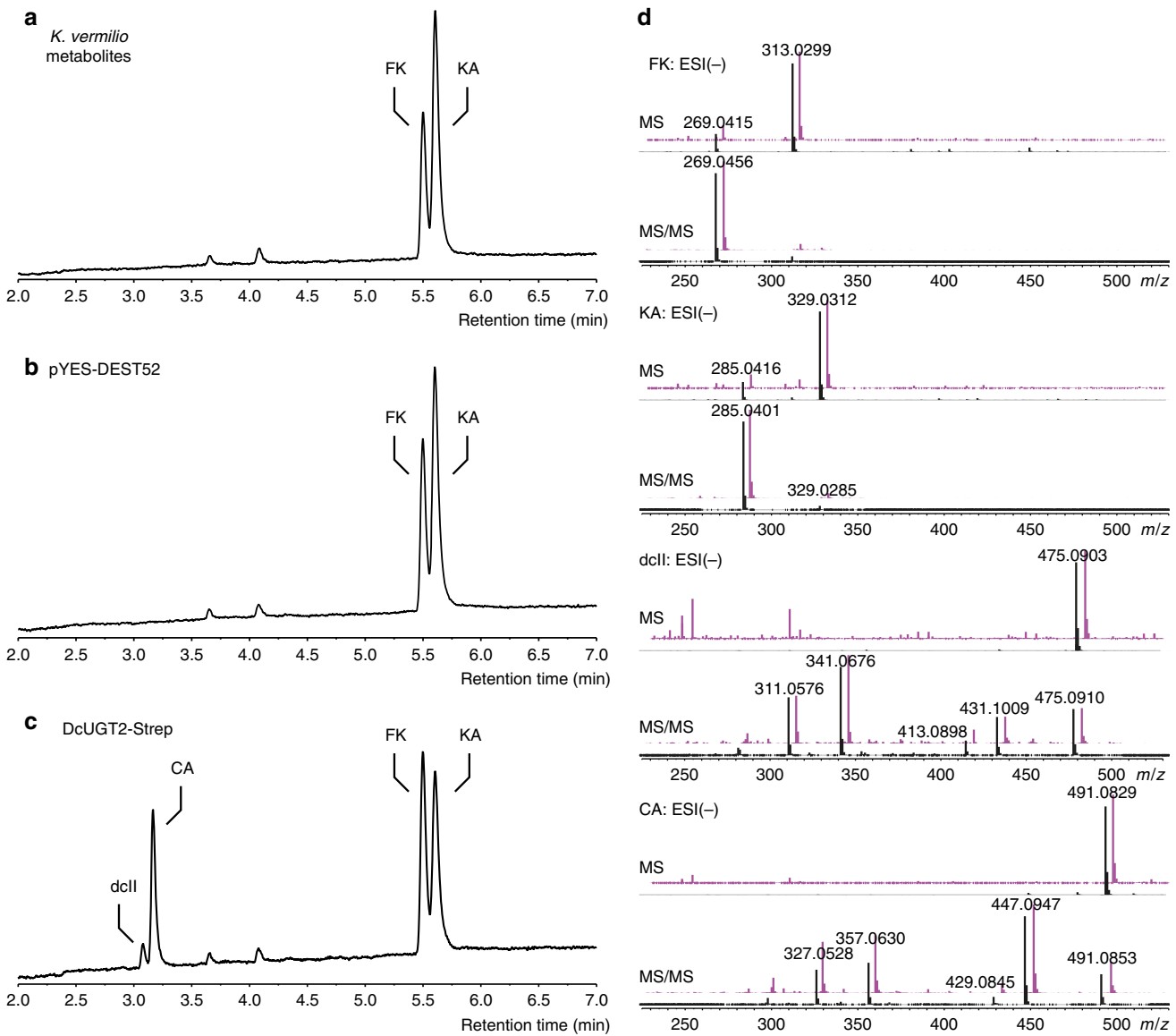

**Fig. 5** DcUGT2 catalyzes CA and dcII formation in vitro. Yeast microsomes containing (**b**) pYES-DEST52 vector or (**c**) DcUGT2-Strep were incubated with UDP-glucose and a *Kermes vermilio* metabolite fraction, containing flavokermesic acid (FK) and kermesic acid (KA) (shown in **a**). Assay products were analyzed by HPLC-HRMS/MS. Peaks indicated in the MS base peak chromatograms (**a–c**) were identified by comparing retention times and mass fragmentation spectra. Mass spectra of FK, KA, dcII, and carminic acid (CA) identified in assays (**d**, shown in purple) stacked with those of authenticated standards (**d**, shown in black)

mediated by a transmembrane helix positioned in its C-terminal region and encompassing amino acid residues 469–492 (Fig. 7). At the N terminal, the first approximately 20 amino acids are predicted to constitute a cleavable signal peptide which targets the protein to the ER (Fig. 7). DcUGT2 is presumed to be anchored to the ER membrane with the N-terminal part facing the ER lumen and a short C-terminal part exposed to the cytoplasm. This fits with the membrane orientation of most ER-bound UGTs which are classified as type I transmembrane proteins[20].

With the globular part of the DcUGT2 protein predicted to be in the ER lumen, an in silico search for potential N-glycosylation sites was carried out. The search showed that the enzyme contains three putative asparagines from which N-linkages to sugars may be established (Fig. 7).

To address whether the native DcUGT2 is a glycoprotein, a deglycosylation assay was performed on *D. coccus* microsomes followed by SDS-PAGE and western blot analysis using a rabbit polyclonal peptide antibody recognizing the native DcUGT2. In the untreated microsomes, a single immunoreactive protein with an apparent mass of 75 kDa was detected, suggesting that DcUGT2 might be heavily glycosylated (Fig. 8a). As expected, this immunoreactive protein was not detected in the *D. coccus* soluble protein fraction further supporting the notion that DcUGT2 is ER-bound (Fig. 8a).

After treatment of the *D. coccus* microsomes with deglycosylating enzymes over different periods of time, a prominent immunoreactive protein emerged with an apparent mass of 49 kDa (Fig. 8b). This protein was assumed to be the fully deglycosylated version of the native DcUGT2, although it did not migrate on the SDS-PAGE as a protein with a mass of 57 kDa, the calculated mass based on its amino acid sequence. Such mass discrepancies are not uncommon for membrane proteins[21]. It may be noticed that even after 24 h of treatment, glycosylated DcUGT2 protein remained present in the microsomes (Fig. 8b).

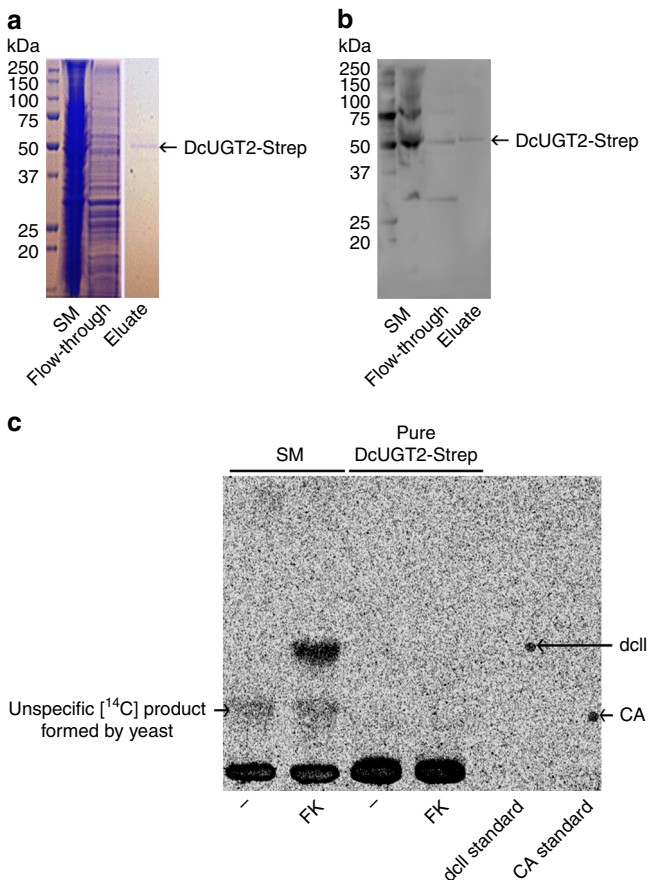

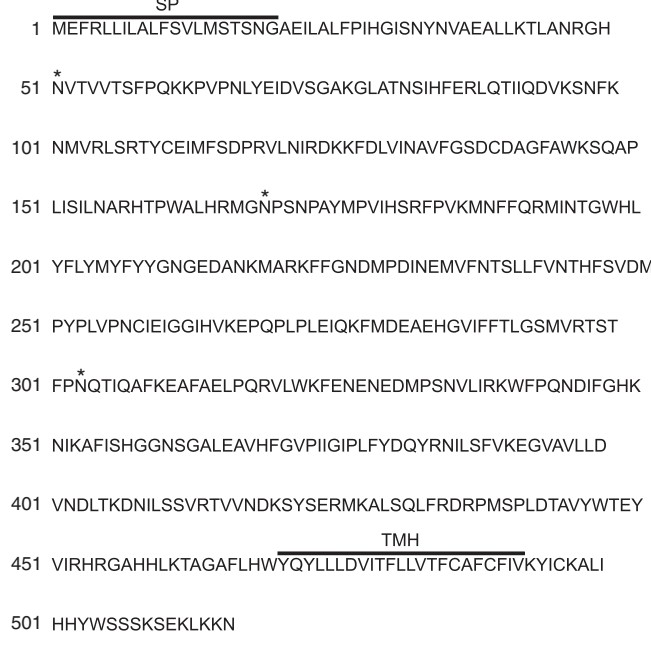

**Fig. 7** DcUGT2 protein sequence. Protein transmembrane topology and signal peptide were predicted by the Phobius sever (accessible at http://phobius.binf.ku.dk). N-glycosylation sites were predicted by the NetNGlyc 1.0 server (accessible at http://cbs.dtu.dk/services/NetNGlyc/). TMH putative transmembrane helix; SP putative cleavable signal peptide; asparagine (N) residues are indicated with an asterisk: potential N-glycosylation sites

**Fig. 6** DcUGT2-Strep loses enzyme activity upon affinity purification. Yeast microsomes containing DcUGT2-Strep were solubilized with reduced Triton X-100 and DcUGT2 affinity purified by its Strep-tag II. Protein fractions were tested for glucosylation activity by incubation with [$^{14}$C] UDP-glucose and with/without flavokermesic acid. **a** Protein samples separated on an SDS gel followed by Coomassie staining. **b** Protein samples separated on an SDS gel followed by western blotting using an anti-Strep antibody. **c** TLC-separated [$^{14}$C] products formed in vitro and viewed by phosphorimaging. SM solubilized microsomes; flow-through proteins that did not bind to the affinity matrix; eluate protein fraction C1 (Supplementary Fig. 3) which was eluted from the affinity matrix with desthiobiotin. FK flavokermesic acid, CA carminic acid. Note that the anti-Strep antibody reacts with the marker protein bands

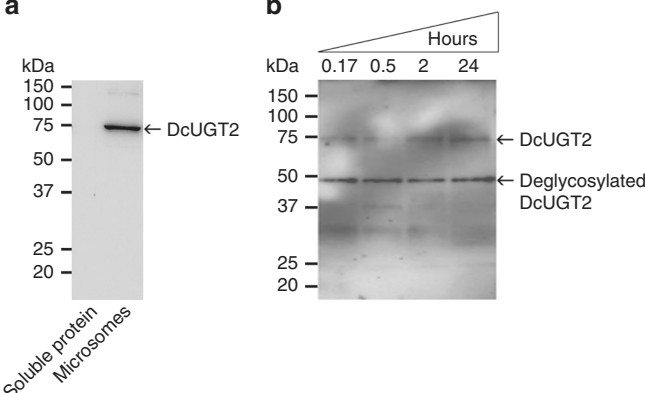

**Fig. 8** The DcUGT2 from *Dactylopius coccus* is a glycoprotein. *D. coccus* proteins were either non-treated or deglycosylated with glycanases for different periods of time. Protein samples were separated on an SDS gel followed by western blotting using an anti-DcUGT2 antibody. **a** Non-treated *D. coccus* soluble protein and microsomes. **b** *D. coccus* microsomes treated with glycanases for 10 min (0.17), 30 min (0.5), 2 h (2), and 24 h (24)

This observation likely reflects that some of the sugar chains, linked to DcUGT2, were less exposed.

In comparison to the sequences available in GenBank provided by the National Center for Biotechnology Information[22], the closest homologous sequence to DcUGT2 is a predicted UDP-glucuronosyltransferase 2B10 from the pea aphid, *Acyrthosiphon pisum*. This UGT shares 46% amino acid sequence identity to DcUGT2 (Supplementary Fig. 4). It is noteworthy that when DcUGT2 was compared to the other 30 putative UGTs annotated in the *D. coccus* transcriptome, the closest homologous sequence was DcUGT23 with a shared amino acid identity of 57%. The assembled transcript encoding *DcUGT23*, however, lacks a stop codon, but the translated product of this partial transcript is 515 amino acids and thus assumed to be nearly full-length.

The alignment of DcUGT2 to several other membrane-bound UGTs shows that the enzyme contains a region between amino acids 46 and 56 which corresponds to the conserved hydrophobic motif "LX2-RG-H-X3-VL", e.g., as described in human UGT1A6[23]. The "LX2-RG-H-X3-VL" sequence region in

DcUGT2 is 91% identical at the amino acid level to the corresponding sequence in UGT2B7 from humans (Supplementary Fig. 4). This motif is critical for the functional and structural integrity of membrane-bound UGTs[20]. Donor binding regions 1 and 2[24] were also identified in DcUGT2 showing 75% and 57% sequence identity, respectively, to those in UGT2B7 (Supplementary Fig. 4). DcUGT2 was also found to contain the important catalytic asparagine residue at position 128. In the sequence alignment with other membrane-bound UGTs, a histidine residue

is predominantly observed as a catalytic residue at position 34[20,24] (Supplementary Fig. 4). In DcUGT2 an asparagine residue is found at position 34.

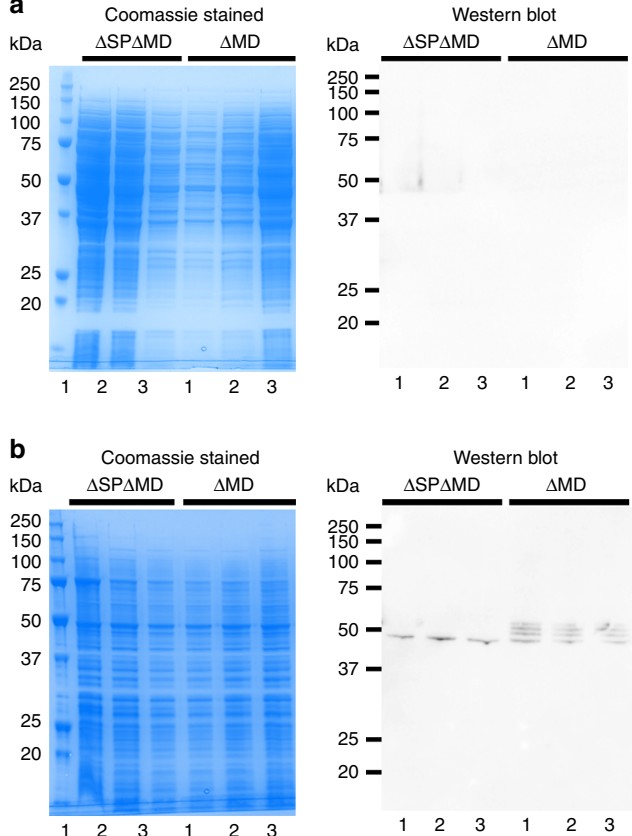

**Fig. 9** Heterologous expression of truncated *DcUGT2* forms in yeast. Three individual *S. cerevisiae* transformants were cultured for each Strep-tagged truncated *DcUGT2* construct, *ΔSP-DcUGT2ΔMD-Strep* and *DcUGT2ΔMD-Strep*. After galactose-induced heterologous protein expression, microsomal and soluble proteins were isolated and separated on an SDS gel followed by either Coomassie staining or western blotting using an anti-Strep antibody. **a** Soluble proteins. **b** Microsomal proteins. ΔSPΔMD ΔSP-DcUGT2ΔMD-Strep, ΔMD DcUGT2ΔMD-Strep

**Targeting DcUGT2 to the ER is critical for its activity**. The predicted localization of the main globular part of DcUGT2 in the ER lumen and its potential N-glycosylation prompted us to investigate whether such compartmentalization would impact its catalytic activity. To test this, two truncated Strep-tagged versions of *DcUGT2* were generated and heterologously expressed in yeast. *DcUGT2ΔMD-Strep* lacked the predicted transmembrane domain and associated cytoplasmic tail (amino acid residues 469–515) while *ΔSP-DcUGT2ΔMD-Strep* lacked the putative N-terminal signal peptide (amino acid residues 1–20) as well as the predicted transmembrane domain and cytoplasmic tail (amino acid residues 469–515). Western blot analyses of both the soluble protein and membrane-bound protein fraction isolated from yeast cultures expressing the truncated DcUGT2 proteins showed that they were synthesized successfully (Fig. 9). The DcUGT2ΔMD-Strep protein was only present in the microsomal fraction. Four distinct immunoreactive proteins were detected ranging in masses from approximately 47 to 58 kDa (Supplementary Fig. 5). Following deglycosylation, the proteins with masses between 50 and 58 kDa disappeared, whereas the immunoreactive protein with the mass of 47 kDa became more prominent and thus most likely represented fully deglycosylated DcUGT2ΔMD-Strep. Interestingly, the truncated ΔSP-DcUGT2ΔMD-Strep also appeared in the microsomal fraction, suggesting that the DcUGT2 protein, apart from the signal peptide, might contain other internal amino acid regions that target the protein to the ER (Fig. 9). It should be noted that the ΔSP-DcUGT2ΔMD-Strep did not appear to be glycosylated and therefore most likely never entered the ER lumen but rather was associated with the ER membrane facing the cytosol or other cellular membrane structures.

In vitro activity assays using yeast microsomes containing the full-length DcUGT2-Strep, truncated DcUGT2-Strep versions, or microsomes from yeast harboring the pYES-DEST52 vector showed that only DcUGT2 versions expressed with the N-terminal signal peptide were catalytically active when compared to the vector control (Fig. 10). Although functionally active, the truncated DcUGT2ΔMD-Strep was not as efficient as DcUGT2-Strep. The production of CA was reduced by two orders of magnitude and production of dcII was abolished when compared with DcUGT2-Strep (Fig. 10). In contrast, the ΔSP-DcUGT2ΔMD-Strep was completely inactive. We conclude that targeting of DcUGT2 to the ER lumen is critical for its activity. The transmembrane domain/cytoplasmic tail are also important to gain optimal activity but are not crucial.

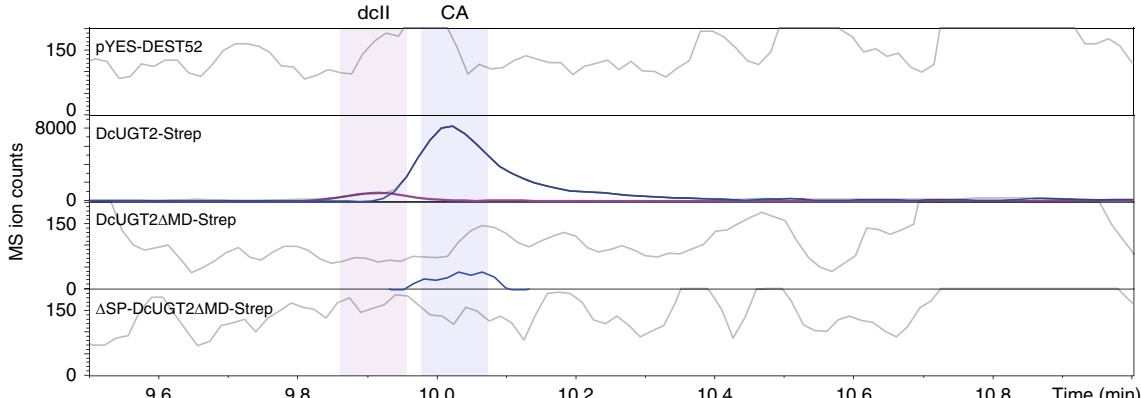

**Fig. 10** Activities of different DcUGT2-Strep forms synthesized in yeast. Yeast microsomes containing pYES-DEST52, DcUGT2-Strep, DcUGT2ΔMD-Strep and ΔSP-DcUGT2ΔMD-Strep were incubated with UDP-glucose and a *Kermes vermilio* metabolite fraction, containing kermesic acid and flavokermesic acid. Assay products were analyzed by LC-ESI(−)-MS and ion chromatograms of *m/z* 475.0882 (purple) and 491.0831 (blue) were extracted, corresponding to dcII and carminic acid (CA), respectively

## Discussion

Although the CA pigment has served as an important red colorant throughout history and its origin from cochineal is well established, no genetic or biochemical information is currently known about its biosynthesis. Due to the chemical structure of CA, the general consensus is that a PKS enzyme is involved in its formation[10]. The present lack of such biochemically identified enzymes from animals has raised speculations as to whether an endosymbiont might be responsible for the CA biosynthesis in cochineals[25]. Several *Dactylopius* species have been shown to contain a multitude of endosymbiotic bacteria, but whether any of these organisms are capable of producing CA remain uncertain[10,26,27]. In *Dactylopius*, the CA pigment is found throughout the body of the insect with very high amounts appearing in the hemolymph[3,28]. Thus, it has been proposed that specialized hemocyte cells, which occur in the hemolymph and have special biosynthetic and secretory function, might be responsible for the production of CA[29–31]. In the current study, a membrane-bound C-glucosyltransferase, DcUGT2, from *D. coccus* has been isolated which is capable of forming CA by glucosylation of KA in vitro. Thus, it is likely that part, if not all, of the biosynthetic pathway leading to the formation of CA is performed by the cochineal insect itself. Several taxonomically widespread dye-producing scale insect species have been shown to contain pigments derived from an FK backbone, indicating that the pathway for FK synthesis has emerged from a common ancestor. As the origin of the anthraquinone backbone of CA is unresolved, the possibility still exists that the KA aglucone may arise from an endosymbiont. In this case, the C-glucosylation by the cochineal would then be considered to be an action of detoxification[32]. Generally, glycosylation serves to stabilize labile aglycons, to increase their solubility, facilitate compartmentalized storage, and to reduce their bioactivity/autotoxicity. This is the reason why many plant defense compounds are stored as glucosides. In some cases, the sugar moiety is cleaved off to activate and jack-up the efficacy of the defense system upon demand. In *D. coccus*, the C-glucosylation step would be expected to facilitate transport, packing, and safe storage of CA. CA is envisioned to serve as a defense compound due to its feeding-deterrent properties towards ants[3]. Storage of toxic constituents is a challenge that not only *D. coccus* but all organisms need to handle and master if they want to use them as part of their defense systems towards predators and pests.

DcUGT2 is predicted to be a type I integral ER membrane protein. In accordance, it possesses a putative N-terminal cleavable signal peptide and a potential C-terminal transmembrane helix, which enable embedment of the enzyme into the ER membrane, with the globular part residing in the ER lumen and a short part exposed to the cytosol. The presence of the putative signal peptide is essential for obtaining a functional active DcUGT2 protein as demonstrated by complete obliteration of glucosylation activity of the truncated DcUGT2 protein devoid of the putative signal peptide and transmembrane domain/cytoplasmic tail, whereas truncated DcUGT2 protein missing only the transmembrane domain/cytoplasmic tail retained activity. The yeast-produced ΔSP-DcUGT2ΔMD-Strep protein remained associated with the membrane protein fraction, indicating that it had been targeted to the ER or associated with other cellular membrane structures in the yeast. If the truncated DcUGT2 protein is indeed targeted to the ER without the putative signal peptide, another unknown ER-targeting signal must be present in the DcUGT2 protein and clearly not contained within the transmembrane domain/cytoplasmic tail. Such a signal has, in fact, been demonstrated to occur within an amino acid stretch, encompassing residues 140–240 of the human UGT1A6, although an exact motif was not defined[33,34]. Expression of *UGT1A6*

without the sequence encoding the N-terminal signal peptide showed that the ΔSP-UGT1A6 protein was translocated into and retained in the ER via this 100 amino acid stretch in mammalian cells[33,34]. The amino acid alignment of DcUGT2 to UGT1A6 indeed identified regions of homology between the two proteins in this 100 amino acid stretch, but whether the ER-targeting signal is contained within these regions remains to be established (Supplementary Fig. 4). In addition to being N-glycosylated in an in vitro transcription–translation system with pancreatic microsomal membranes and in vivo when expressed in the *Pichia pastoris* yeast, the ΔSP-UGT1A6 protein was also functionally active and had similar kinetic parameters to UGT1A6[33,34]. It was therefore concluded that the signal peptide was not essential for membrane assembly and functional activity of UGT1A6. The authors also found that when the signal peptide and transmembrane domain/cytoplasmic tail were removed, the truncated UGT1A6 protein remained able to enter the ER and undergo N-glycosylation in *P. pastoris*. In contrast, the yeast-produced ΔSP-DcUGT2ΔMD-Strep protein, although associated with the membrane fraction, did not seem to enter into the lumen of the ER, as indicated by the lack of post-translational N-glycosylation. The native *D. coccus* DcUGT2 was shown to be subject to heavy glycosylation. Whether such post-translational modifications are required for its catalytic activity is uncertain. Physiological conditions like the redox potential of the environment and protein factors including chaperones present in the ER may be essential for the glucosylation mechanism of DcUGT2. A fair amount of evidence point to the functioning of membrane-bound UGTs as dimers or oligomers in vivo. Proper membrane integration may be a prerequisite for efficient assembly of the UGT monomers[20,35]. The dimerization/oligomerization process has been proposed to greatly increase the metabolic capacity of membrane-bound UGTs[36]. Hampered ability of the truncated DcUGT2 to oligomerize might thus affect the enzyme activity negatively. It is noteworthy that the yeast-synthesized DcUGT2ΔMD-Strep protein appeared as distinctly glycosylated as well as non-glycosylated variants after expression. This could indicate that some DcUGT2ΔMD-Strep molecules never entered into the ER, while others entered and were positioned differently along the secretory pathway where they encountered different N-glycosyltransferases. Thus, the impaired enzyme activity might simply reflect that the heterologously produced DcUGT2ΔMD-Strep was a mixed population of glycosylated and non-glycosylated variants. Based on these findings we conclude that targeting DcUGT2 to the lumen of the ER is essential for the functional activity.

The truncations of the N-terminal signal peptide and of the transmembrane domain/cytoplasmic tail were initially designed to generate a soluble functional variant of DcUGT2 suitable for heterologous production of CA in a prokaryotic host. In light of the results obtained in the current study, it is reasoned that it might be necessary to use bacterial strains engineered to perform post-translational glycosylation and mimic ER conditions in order to produce CA. Alternative platforms based on the use of eukaryotic organisms such as yeasts or algae are likely to be more suitable for CA production.

## Methods

**Transcriptomic analysis**. Frozen adult female *D. coccus* (0.5 mg) obtained from Lanzarote, Spain were ground into a fine powder with a mortar and pestle under liquid nitrogen. Total RNA was subsequently extracted using the RNeasy Mini Kit (Qiagen) according to the manufacturer's instructions. Polyadenylated RNA was converted into cDNA with an oligo-dT primer and a reverse transcriptase (RT$^2$ Easy First Strand Kit, Qiagen). The cDNA samples were sequenced with a total yield of 5 GB sample$^{-1}$ (corresponding to 51 million 90-bp reads) by BGI-Shenzhen, China using 90-bp paired-end Illumina sequencing technology. Sequenced paired-end reads were assembled de novo into contigs using the Genomic Workbench version 5.4 software (CLC bio, Qiagen). Quality-based read trimming

was performed based on Phred scores, using a modified Mott-trimming algorithm with a limit of 0.05 and a maximum of 2 ambiguous bases/reads after trimming. More details about the Mott-trimming algorithm used by CLC bio can be found in online documentation[37]. De novo transcriptome assembly was carried out using the de Bruijn graph algorithm in the CLC bio Genomic Workbench. Settings were a word size of 20, a bubble size of 50, and a minimum contig length of 200. After assembly, the reads were mapped back to the contigs with the following mapping parameters: mismatch cost = 2, insertion cost = 3, deletion cost = 3, length fraction = 0.5, and similarity fraction = 0.8. Putative genes were identified using the hidden Markov matrix-based prokaryote gene finder tool in IOGMA v. 10 (Genostar, Grenoble, France). This approach was regarded to be more simple than using the eukaryote gene finder tool since only polyadenylated RNA, in which splicing events are presumed to already have occurred, was analyzed. Annotation of putative UGT genes was carried using both the nucleotide and translated protein sequences in a BLAST comparison with the GenBank sequence database (National Center for Biotechnology Information, NCBI) and by similarity comparison to the UDPGT (UDP-glucuronosyl and UDP-glucosyltransferase) Pfam protein family (PF00201)[38].

**Preparation of protein fractions.** Fresh *D. coccus* insects (3 g) were homogenized in 120 ml of isolation buffer (350 mM sucrose, 20 mM Tricine (pH 7.9), 10 mM NaCl, 5 mM DTT, 1 mM PMSF, Complete protease inhibitor cocktail tablets (Roche) containing 0.3 g polyvinylpolypyrrolidone). The homogenate was filtered through a nylon cloth (22 μm mesh) and centrifuged (10 min, 10,000×g, 4 °C). The supernatant was isolated and ultracentrifuged (1 h, 105,000×g, 4 °C), yielding a soluble and a membrane-bound protein fraction. The soluble protein fraction was concentrated to 1 ml and buffer-exchanged with 20 mM Tricine (pH 7.9), and 5 mM DTT by using Amicon Ultra centrifugal filter-3K devices (Millipore). The membrane-bound protein pellet was washed thrice by resuspending the pellet in 60 ml of 20 mM Tricine (pH 7.9), and 5 mM DTT using a marten paintbrush followed by ultracentrifugation. The membrane-bound protein pellet was finally resuspended in 1 ml of 20 mM Tricine (pH 7.9), and 5 mM DTT. The soluble protein fraction and the membrane-bound protein fraction were analyzed for glucosylation activity.

**LC-MS/MS analysis of protein fractions with UGT activity.** The membrane-bound protein fraction isolated from fresh *D. coccus* insects (10 g), as described above, was solubilized by adding reduced Triton X-100 to a final concentration of 1% (v v$^{-1}$), gently stirred (1.5 h, 4 °C), and centrifuged (1 h, 105,000×g, 4 °C). The supernatant was isolated and applied to a column packed with 2 ml Q-Sepharose Fast flow (GE Healthcare). The column was washed in 4 ml of buffer A (20 mM Tricine (pH 7.9), 0.1% (v v$^{-1}$) reduced Triton X-100, 50 mM NaCl) and proteins were eluted with 20 mM Tricine (pH 7.9) and 0.1% (v v$^{-1}$) reduced Triton X-100 using a stepwise NaCl gradient from 100 to 500 mM with 50 mM increments. Fractions (0.5 ml) were collected, desalted, analyzed by SDS-PAGE, and monitored for glucosylation activity using the described [$^{14}$C]glucosylation enzyme assay. A fraction showing increased FK/KA-specific UGT activity was separated on a 12% SDS gel and two gel blocks spanning the 50–70 kDa region were excised. The gel blocks were digested with trypsin after reduction and alkylation according to Shevchenko et al.[39] and eluted with 0.1% trifluoroacetic acid. LC-MS/MS: reverse phase nano-HPLC was coupled online to a tandem LTQ-orbitrap XL electrospray mass spectrometer: Chromatographic separation was performed by an EASY-nLC system (Thermo, Bremen, Germany). The peptides were separated by a two column system that consisted of a 2 cm trap column of ReproSil-Pur 120 AQ-C18, 3 μm (Dr Maisch GmbH, Ammerbuch Entringen, Germany) packed in 100 μm fused silica fitted with a kasil plug and connected to the separation column which was packed to 10 cm in a 75 μm pulled needle fused silica capillary with the same material as in the trap column. After loading and desalting, the peptides were separated with a linear gradient from 0 to 32% in solvent B in 60 min and 32 to 100% in solvent B in 5 min at a flow rate of 250 nl min$^{-1}$. Solvent A was composed of 0.1% formic acid in water and solvent B was composed of 95% acetonitrile, 0.1% formic acid, and 5% water. Mass spectra were acquired in the positive ion mode. Settings were as follows: The electrospray voltage was kept at 2.3 kV with an ion transfer temperature of 270 °C with active background ion reduction (New Objective Inc., Woburn MA, USA) gas flow. Data-dependent acquisition was used for automated switching between MS mode in the orbitrap and MS/MS mode in the LTQ. Charges of 1,000,000 were accumulated in the LTQ before injection in the orbitrap in which a parent ion scan from *m/z* 300–1800 was performed with a target peak resolution of 60,000 at *m/z* 400. The five most abundant ions with charge states above 1 and intensity above 15,000 counts were selected with an isolation width of 2.5 *m/z* units for MS/MS with collision-induced dissociation in the LTQ. Charges of 30,000 were accumulated, the normalized collision energy was set to 35% with activation q = 0.25 and activation time 30 ms. *m/z* values ±10 p.p. m. of precursor ions that were selected for MS/MS were subjected to a dynamic exclusion list for 45 s. LC-MS/MS data were searched with a MASCOT server (Matrix Science) operated by Proteome Discoverer software (Thermo Scientific) against the de novo-assembled *D. coccus* transcriptome database. Carbamido-methyl was set as fixed modification and deamidation of asparagine and glutamine residues and oxidation of methionine residues as variable modifications. The peptide MS and MS/MS tolerances were set to 10 p.p.m. and 0.8 Da, respectively.

The Decoy database was searched for peptide false discovery rate determination. The expected value was adjusted to match a strict false discovery rate of 1% by the Target Decoy PSM Validator module of Proteome Discoverer. At least two peptides were required for identification. Identified protein sequences from the de novo-assembled *D. coccus* transcriptome database were subjected to BLAST search against insect proteins for functional annotation.

**Cloning of *DcUGT* fragments and yeast heterologous expression.** Full-length *DcUGT* candidates (*DcUGT2*, *DcUGT4*, and *DcUGT5*) were either verified or obtained by rapid amplification of cDNA ends from polyadenylated RNA of adult female *D. coccus* by using the SMARTer RACE 5′/3′ Kit (Clontech). The three cDNAs and the following constructs thereof were sequenced by Macrogen Inc. The candidate *DcUGTs* were codon optimized for *S. cerevisiae* expression and synthesized with Gateway-compatible attL recombination sites by GenScript. The synthetic genes were used as templates with specific primers in sequential PCRs to generate the corresponding Strep-tagged versions. In the first PCR, the candidates were amplified with the forward primer, attB1: 5′-GGGGACAAGTTTGTA-CAAAAAAGCAGGCT-3′ and a specific reverse primer. Specific reverse primers were: 5′-TTATTTTTCGAATTGTGGATGAGACCAAGCAGAATTCTTTTTC AACTTTTCAGATTTAG-3′ (*DcUGT2*), 5′-TTATTTTTCGAATTGTGGATGA-GACCAAGCAGATTTTGTTAACATTCTGAAAAAGATTCT-3′ (*DcUGT4*), and 5′-TTATTTTTCGAATTGTGGATGAGACCAAGCAGAGTTATCCTTAACT TTCTTAGTTGGTTT-3′ (*DcUGT5*). The truncated versions, *DcUGT2ΔMD-Strep* lacking the predicted transmembrane domain/cytoplasmic tail and *ΔSP-DcUGT2ΔMD-Strep* lacking the putative N-terminal signal peptide and the predicted transmembrane domain/cytoplasmic tail, were amplified from the synthetic *DcUGT2* gene. Primer sets used in the first PCR were: attB1/MD-Strep: 5′-TTAT TTTTCGAATTGTGGATGAGACCAAGCAGAGTGCAAAAAGGCACCTG CAGT-3′ for the amplification of *DcUGT2ΔMD-Strep* and 5′-CAAGTTTGTA-CAAAAAAGCAGGCTAAAAATGCCGAAATCTTGGCTTTATTCC-3′/MD-Strep for the amplification of *ΔSP-DcUGT2ΔMD-Strep*. All products from first PCR were diluted 15 times and used in a second PCR with the forward attB1 primer and a reverse primer: Strep_attB2: 5′-GGGGACCACTTTGTACAAGAAAGCTGGGT CTTATTTTTCGAATTGTGGATGAGAC-3′, resulting in C-terminal Strep-tagged fragments flanked by Gateway-compatible attB sites. These fragments were cloned into pDONR207 (Invitrogen) and then transferred into destination vector, *pYES-DEST52* (Invitrogen), using Gateway Technology (Invitrogen) according to the manufacturer's instructions. Recombinant pYES-DEST52 constructs and *pYES-DEST52* were separately transformed into the Invsc1 yeast strain (Invitrogen) and positive transformants were verified by PCR. Heterologous protein production was carried out as described in the *pYES-DEST52* manual (Invitrogen). Soluble proteins and membrane-bound proteins (microsomes) were isolated according to Pompon et al.[40]. Yeast cells were harvested from 25-ml cultures by centrifugation (10 min, 7500×g, 4 °C) and washed with 1 ml TEK buffer (50 mM Tris-HCl (pH 7.5), 1 mM ethylenediaminetetraacetic acid (EDTA) and 100 mM KCl). The cells were sedimented by centrifugation (10 min, 7500×g, 4 °C) followed by resuspension in 1 ml TES2 buffer (50 mM Tris-HCl (pH 7.5), 1 mM EDTA and 600 mM sorbitol, 1% (w v$^{-1}$) bovine serum albumin, 5 mM DTT, and 1 mM PMSF). Yeast cell disruption was achieved by a 5-min votexing with acid-washed glass beads (425–600 μm; Sigma-Aldrich) at 4 °C. The supernatant was collected by centrifugation (15 min, 10,000×g, 4 °C) and ultracentrifuged (1 h, 105,000×g, 4 °C), yielding a soluble protein fraction and a microsomal pellet. The microsomal pellet was subsequently washed twice by resuspending the pellet in 5 ml TES buffer (50 mM Tris-HCl (pH 7.5), 1 mM EDTA, and 600 mM sorbitol) and once in TEG buffer (50 mM Tris-HCl (pH 7.5), 1 mM EDTA, and 30% (v v$^{-1}$) glycerol) using a marten paintbrush followed by ultracentrifugation in between. The membrane-bound protein pellet was finally resuspended in 0.5 ml of TEG buffer. Production of heterologous Strep-tagged protein was verified by western blotting using an anti-Strep antibody (Qiagen, catalog no. 34850; in a 1:2000 dilution) followed by a secondary horse-radish peroxidase (HRP)-conjugated antibody (Pierce Biotechnology, catalog no. 1858413; in a dilution of 1:5000) and chemiluminescence detection.

**Affinity purification of Strep-tagged DcUGT2.** Yeast microsomes containing Strep-tagged DcUGT2 were isolated from a 250 ml culture, resuspended in 30 ml of binding buffer (100 mM Tris-HCl (pH 7.5), 150 mM NaCl, and 1 mM EDTA), and solubilized by adding reduced Triton X-100 to a final concentration of 1% (v v$^{-1}$) under gentle stirring (1.5 h, 4 °C). The supernatant was isolated by ultra-centrifugation (1 h, 105,000×g, 4 °C) and the Strep-tagged DcUGT2 affinity purified on an equilibrated 5 ml *Strep*-Tactin column (IBA GmbH), operated by an ÄKTA explorer 100 FPLC system (GE Life Sciences) and a flow rate of 1 ml min$^{-1}$. Column equilibration and washing were according to the manufacturer's guidelines. Protein elution was carried out with a flow rate of 3 ml min$^{-1}$ using 10 column volumes of binding buffer containing 2.5 mM desthiobiotin in a gradient of 0–100%. Fractions (0.5 ml) were collected, desalted, analyzed by SDS-PAGE, and monitored for glucosylation activity using the described [$^{14}$C]glucosylation enzyme assay.

**Enzyme assays and glucoside product detection.** Assays were carried out using either UDP-glucose or [$^{14}$C]UDP-glucose as the sugar donor. [$^{14}$C] assays were

performed in reaction mixtures (total volume: 60 µl) containing 20 mM Tricine (pH 7.9), 0.2 mM aglucone substrate, 3.3 µM [$^{14}$C]UDP-glucose (specific activity: 302 Ci mmol$^{-1}$), and 20 µl protein extract (membrane-bound or soluble protein) in a final concentration of 0.5 mg ml$^{-1}$. Following incubation (0.5 h, 30 °C), the reactions were terminated by adding 180 µl of methanol. Samples were centrifuged (5 min, 16,000×$g$, 4 °C) and supernatant was applied to TLC plates (silica gel 60 F254 plates; Merck). Radiolabeled products formed were resolved in dichloromethane:methanol:formic acid (7:2:2, by volume). [$^{14}$C]-labeled products were visualized using a STORM 840 PhosphorImager (Molecular Dynamics). Non-radioactive assays were performed in reaction mixtures (total volume: 60 µl) containing 20 mM Tricine (pH 7.9), kermes metabolite extract (containing both FK/KA), 1.25 mM UDP-glucose, and 20 µl protein extract (membrane-bound or soluble protein). After incubation (2 h, 30 °C), the reactions were terminated by adding 180 µl of methanol and passed through a 0.45 µM hydrophilic low protein binding spin filter (Millipore). Assay products were detected using two different LC-HRMS systems. System 1 consisted of an Agilent 1290 HPLC (Santa Clara, CA, USA), which include a binary pump, a thermostatically controlled column compartment maintained at 35 °C, equipped with a Kinetix XB-C18 column (100 mm × 4.60 mm, 2.6 µm, 100 Å; Phenomenex, Torrance, CA, USA) and a photodiode-array detector, connected to an Agilent Q-TOF equipped with an electrospray ionization source operated in negative ionization mode. Separation was obtained using gradient elution of water–methanol (75:25) (eluent A) and methanol–water (70:30) (eluent B), both acidified with 5% formic acid. The following elution profile was used, with a flow rate of 0.8 ml min$^{-1}$ operated at 35 °C: 0–0.5 min, 100% A; 1.5 min, 69% A; 2.5 min, 37% A; 4.5 min, 13% A; 10 min, 0% A; 11 min, 0% A; 12.5 min, 100% A. Retention times were 3.1 min for dcII, 3.2 min for CA, 5.5 min for FK, and 5.6 min for KA. System 2 consisted of an Agilent 1260 series HPLC system comprising a G1311B quaternary pump with built-in degasser, a G1329B autosampler, a G1316A thermostatically controlled column compartment, and a G1315D photodiode-array detector connected to a Bruker micrOTOF-Q II (Bruker Daltonics Inc., Billerica, MA, USA) equipped with an electrospray ionization source operated in negative ionization mode. Chromatographic separation was performed at 40 °C on a Phenomenex Luna C$_{18}$(2) column (150 × 4.6 mm$^2$, 3 µm, 100 Å), using water–acetonitrile (95:5) (eluent A) and acetonitrile–water (95:5) (eluent B), both acidified with 0.1% formic acid. The following gradient elution profile was used at a flow rate of 0.8 ml min$^{-1}$: 0 min, 100% A; 20 min, 0% A, 22 min, 0% A; 24 min, 100% A. On system 2, retention times were 9.9 min for dcII, 10.0 min for CA, 14.7 min for FK, and 14.8 min for KA.

**Viscozyme treatment of [$^{14}$C]-labeled products**. [$^{14}$C]-labeled products, formed in in vitro enzyme assays, were dried completely under a nitrogen gas flow and resuspended in 30 µl of 50 mM citrate buffer (pH 4.7). Following addition of 1 µl of Viscozyme L (0.121 Fungal Beta-Glucanase units; Novozymes), samples were incubated (3 h, 55 °C) and reactions terminated by adding 90 µl of methanol. The Viscozyme-treated samples were separated by TLC using the solvent system dichloromethane:methanol:formic acid (7:2:2, by volume). [$^{14}$C]-labeled products were visualized by phosphorimaging. [$^{14}$C]Linamarin was produced enzymatically by using a recombinant S-tagged cassava UGT, UGT85K4 (accession no. AEO45781) synthesized in *Escherichia coli*. Crude *E. coli* lysate containing 0.5 µg of S-tagged UGT85K4 was incubated in an assay mixture of 20 µl composed of 100 mM Tris-HCl (pH 7.5), 3.3 µM [$^{14}$C]UDP-glucose (specific activity: 302 Ci mmol$^{-1}$), and 5 mM acetone cyanohydrin. The reaction was incubated (0.5 h, 30 °C) and terminated by adding 2 µl 10% (v v$^{-1}$) acetic acid[41]. The produced [$^{14}$C]Linamarin was dried completely under a nitrogen gas flow prior to Viscozyme treatment.

**Substrates for glucosylation assays**. [$^{14}$C]UDP-glucose supplied by Perkin-Elmer NEN Radiochemicals was dried under nitrogen and then redissolved in 20 mM Tricine (pH 7.9) before use. UDP-glucose was purchased from Sigma-Aldrich. FK and dcII were isolated by extracting dried and ground *D. coccus* with methanol–water (1:1 (v v$^{-1}$)) adjusted to pH 3 with formic acid. The extract was partitioned three times between ethyl acetate and the ethyl acetate phases were collected, combined, and concentrated in vacuo. The extract was then subjected to ion-exchange chromatography using a column packed with Sepra NH$_2$ functionalized silica (Phenomenex). The column was equilibrated in acetonitrile–water (1:1 (v v$^{-1}$)) containing 10 mM ammonium formate prior to application of the extract. Subsequently, the column was washed with the equilibration solvent followed by elution of FK and dcII with acetonitrile–water (1:1 (v v$^{-1}$)) adjusted to pH 11 with ammonium hydroxide. Final isolation was achieved on a column packed with Isolute diol functionalized silica (Biotage), using a stepwise elution gradient from dichloromethane-to-ethyl acetate-to-methanol, to afford FK and dcII. Their molecular structures were verified by one-dimensional and two-dimensional nuclear magnetic resonance. Asperthecin was extracted from *A. nidulans* with ethyl acetate + 1% formic acid. The extract was subjected to flash chromatography on a 10 g diol column (Biotage) and eluted stepwise with dichloromethane, ethyl acetate, and methanol. Final purification of asperthecin was achieved by semipreparative HPLC using a LUNA(2) C$_{18}$ column (Phenomenex) that was eluted with a linear acetonitrile–H$_2$O gradient consisting of A: H$_2$O + 50 p.p.m. trifluoroacetic acid and B: acetonitrile + 50 p.p.m. trifluoroacetic acid from 20 to 60% B over 20 min. The

isolated asperthecin was verified by comparison to an analytical standard where both retention time and accurate mass matched that of the standard. A metabolite fraction containing both KA and FK was isolated from dry *K. vermilio* insects obtained from Kremer Pigmente GmbH & Co. KG (Germany). The extraction of the *K. vermilio* metabolite fraction was carried out using the same method specified above for extracting FK and dcII from *D. coccus*.

**Protein deglycosylation**. Microsomal proteins from either *D. coccus* or yeast synthesizing DcUGT2ΔMD-Strep were deglycosylated using the Enzymatic Deglycosylation Kit for N-linked and Simple O-linked glycans (Prozyme) according to the supplier's instruction. Deglycosylation was monitored by western blot analysis using either an anti-Strep antibody (Qiagen, catalog no. 34850; in a dilution of 1:2000) followed by a secondary HRP-conjugated antibody (Pierce Biotechnology, catalog no. 1858413; in a dilution of 1:5000) or an anti-DcUGT2 antibody (in a 1:1000 dilution) followed by a secondary HRP-conjugated antibody (Dako, catalog no. P0217; in a dilution of 1:5000) and chemiluminescence detection. Blocking, antibody probing, and washing of the blots were performed according to the manufacturers' instructions. Uncropped images are shown in Supplementary Fig. 6. The anti-DcUGT2 antibody was obtained by immunizing a rabbit with the sequence-specific peptide, (NH$_2$)-CEIMFSDPRVLNIRDKKFD-(COOH), representing residues 110–128 in the DcUGT2 protein, conjugated to keyhole limpet hemocyanin (Agisera AB). The pre-immune serum (in a 1:1000 dilution) of the immunized rabbit was tested for cross-reactivity towards a crude *D. coccus* protein extract to ensure the anti-DcUGT2 antibody specificity (Supplementary Fig. 7).

**Data availability**. Raw sequencing reads of the *D. coccus* transcriptome have been submitted to the Sequence Read Archive (SRA) database at National Center for Biotechnology Information as a BioSample (sample accession code SAMN06806158 under experiment accession code SRX2750223). The following cDNA sequences are deposited at the National Center for Biotechnology Information: *DcUGT2* (accession code KY860725), *DcUGT4* (accession code KY860726), and *DcUGT5* (accession code KY860727). The peak-list file used for protein identification in MASCOT searches is given in Supplementary Data 3. All other data are available from the corresponding authors upon reasonable request.

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

## Acknowledgements
This work was supported by a grant from the Danish National Advanced Technology Foundation (now Innovation Fund Denmark) grant no. 018-2011-1 and by the UCPH Excellence Programme for Interdisciplinary research to Center for Synthetic Biology. We thank Cultivo de la cochinilla en Mala y Guatiza for supplying fresh *D. coccus* from Lanzarote, Spain and Dr. Tomas Laursen for valuable advice on membrane protein purification.

## Author contributions
R.K. and B.L.M. wrote the manuscript with contribution from all authors. R.K., B.L.M., F. T.O. and R.J.N.F. provided the overall planning of the project and scientific mentoring and discussion. R.K. planned and designed experiments, purified the DcUGT2 activity from *D. coccus*, constructed yeast strains, and identified and characterized the full-length *D. coccus* DcUGT2 with respect to activity in cold and radiolabeled activity assays. L.S. generated and characterized truncated DcUGT2 proteins, constructed yeast strains, and carried out deglycosylation experiments. J.B.-J. performed the proteomics analysis. M.B. carried out mRNA extraction, transcriptomic sequencing of *D. coccus*, and mined the *D. coccus* transcriptome for putative *UGT* genes. K.T.K. and D.S. carried out the LC-MS analyses. B.M. isolated a metabolite fraction from *Kermes vermilio* containing KA and FK and performed LC-MS analyses. S.A.R. and T.O.L. isolated the FK and dcII compound from *D. coccus* and the asperthecin compound from *Aspergillus*.

## Additional information

**Competing interests:** In the course of this work, R.K., B.M., M.B., and F.T.O. were employed by Chr. Hansen A/S that produces and sells carmine, derived from *D. coccus*, as a food ingredient. R.K., B.M., M.B., F.T.O., R.J.N.F., and B.L.M. have filed a patent application (Publication number: WO2015091843 A1) specifying the identification and use of DcUGT2 in relation to color production. The remaining authors declare no competing financial interests.

