## [Peer Review File · Nature Communications]

Reviewers' comments:

Reviewer #1 (Remarks to the Author):

Kannangara et al combine bioinformatic and extensive biochemical assays to identify and characterize a C-glucosyltransferase from *D. coccus* that can synthesize carminic acid. Overall I found this to be a well carried out set of experiments and a well written manuscript with which I find no major faults. Below I list only a few minor comments on issues that might improve the final paper.

Figure 5: In (d) you show the MS and MS/MS data for the standards, but not the experiment. I would have liked to see the corresponding data from the CA and dcII peaks in (c) for comparison.

107/Supplementary table 1: reporting average coverage is not typically used to say which are the "highest expressed" transcripts. More typically these abundance values are normalized for gene length along the lines of RPKM (reads per kb sequence per million reads) or another related metric, since coverage values for a transcript can be biased by transcript length, splice variants, etc. While I don't think this will change the conclusions significantly, it would improve the presentation

216: Is it more appropriate to say that the assembled DcUGT23 transcript lacked a stop codon, rather than the isolated transcript? I would think the lack of a stop codon more likely represents an assembly or sequencing artifact than a problem with the RNA transcript itself.

221: missing quote after X3-VL

225: Histidine to asparagine is a relatively conservative mutation; it's certainly fine to note the change at a likely catalytically important residue, but can you add a bit of context/interpretation to this statement? Is there anything known about the impact of this type of mutation at that site for function?

251: Should be catalytically active

372: Should be SMARTer

Methods:

I tried to examine the NCBI datasets, but they appear to not be publicly released yet. Thus, I was not able to independently check these data. I trust that the sequences will be released upon publication.

The transcriptome sequencing and assembly methods need more detail. For reproducibility purposes, please provide specifics on the quality filtering/trimming settings as well as assembly algorithm settings beyond just 'defaults' – such defaults are not particularly obvious to others who do not have the commercial package you used. Also, I am not familiar with the IOGMA software; I presume you used a prokaryotic HMM model because the package would otherwise look for splicing sites in the assembled transcriptome? Or is there another reason why a eukaryotic model was not used?

Also, can you specify which relevant Pfam families you were looking for when looking through the assembled contigs?

While you did confirm the full-length sequence by RACE, was there any hint from the RACE clones, or from re-aligning the transcriptome reads to the transcriptome assemblies, that there might

exist other isoforms of DcUGT2?

Reviewer #2 (Remarks to the Author):

Using transcriptomic, proteomic and biochemical approaches, the authors isolated and characterized a *D. coccus* UDP-glucosyltransferase DcUGT2 that catalyzes C-glucosylation of flavokermesic acid and kermesic acid to yield the 7-C-glucoside of flavokermesic acid (dcII) and the industrially relevant colorant carminic acid, respectively. Although the isolation and characterization of the bulk of enzymes (polyketide synthase, oxygenases, etc.) required for carminic acid biosynthesis is still an unresolved challenge, this work represents progress toward fleshing out a carminic acid biosynthetic pathway. Indeed, armed with the knowledge gained in this work, the authors co-opted a plant polyketide synthase, bacterial cyclases and DcUGT2 to produce dcII in *N. benthamiana* in a follow-up effort (Andersen-Ranberg et al., 10.1002/cbic.201700331). However, a detailed examination of DcUGT2 (the focus of this paper) is more appropriate for a specialized journal. Some specific comments and suggestions follow:

1) Figure 1. Malonyl-CoA is incorrectly drawn as Acetoacetyl-CoA.

2) Page 6. "The four full-length DcUGT cDNAs were expressed in *S. cerevisiae* and *Aspergillus nidulans* without codon optimization. No functional UGT activity was obtained as monitored by the absence of changes in the LC-MS and TLC profiles following incubation of soluble and microsomal protein extracts from the transformed heterologous hosts with UDP-glucose or [¹⁴C]-UDP-glucose, respectively, and the putative substrates flavokermesic acid and kermesic acid when compared to incubation with the un-transformed host controls (data not shown)." Why is there a discrepancy between these unsuccessful heterologous expression experiments and later successful ones? Is it the codon optimization? This disagreement is not explicitly discussed.

3) Page 7. "A protein fraction eluted with 100 mM NaCl was separated by SDS-PAGE and the proteins migrating in the 50 to 75 kDa region were subjected to in gel trypsin digestion, LC-MS/MS- based amino acid sequencing of the fragments obtained and database searching (Fig. 2c)." Why was this fraction selected? Fractions eluting at higher NaCl concentrations also have flavokermesic acid and kermesic acid-specific glucosylation activity.

4) Figure 3. To guide the reader, the texts "DcUGT2-Strep", "DcUGT4-Strep" and "DcUGT5-Strep" should be added above their matching gel/blot pair.

5) Page 10. "As observed from the empty vector control, *S. cerevisiae* also possesses endogenous glucosylation activities, yielding a [¹⁴C]-product with a similar R_f value as carminic acid in the applied TLC system (Fig. 4)." Why are the DcUGT5-Strep lanes' unknown [¹⁴C]-product bands elongated like the DcUGT2-Strep lanes' bands and unlike the pYES-DEST52 lanes' bands?

6) Figure 5. As in Figure 3, to guide the reader, the texts "pYES-DEST52" and "DcUGT2-Strep" should be added above their matching chromatograms.

7) Page 12. "To characterize the kinetic properties of DcUGT2, yeast microsomal membranes containing heterologously-produced DcUGT2 were solubilized using reduced Triton X-100 and DcUGT2 was affinity-purified by its Strep-tag II (Fig. 6). In vitro tests of the isolated Strep-tagged DcUGT2 showed that its enzymatic activity was lost upon isolation (Fig. 6)." The glycosylated form of DcUGT2 looks like it is present in the flow-through and not in the eluate. Were in vitro tests performed using the flow-through?

8) Page 18. "In vitro activity assays using yeast microsomes containing the full-length DcUGT2-Strep, truncated DcUGT2-Strep versions or microsomes from yeast harboring the empty vector showed that only DcUGT2 versions expressed with the N-terminal signal peptide were catalytic active when compared to the empty vector control (Fig. 10). Although functionally active, the truncated DcUGT2ΔMD-Strep was not as efficient as DcUGT2-Strep." The production of carminic acid was reduced by two orders of magnitude and production of dcII was abolished. This paragraph does not accurately describe how impaired DcUGT2ΔMD-Strep is relative to DcUGT2-Strep.

9) Figure 10. As in Figures 3 and 5, to guide the reader, the texts "pYES-DEST52", "DcUGT2-

Strep", "DcUGT2ΔMD-Strep" and "ΔSP-DcUGT2ΔMD-Strep" should be added above their matching chromatograms.

10) Page 20. "The native *D. coccus* DcUGT2 was shown to be subject to heavy glycosylation. Whether such post-translational modifications are required for its catalytic activity is uncertain." Could the three putative asparagines for N-glycosylation be mutated to glutamines to test whether this modification is required for catalytic activity?

Reviewer #3 (Remarks to the Author):

Kannanger et al. present a detailed and well-written manuscript on the identification of the final enzyme in the hitherto uncharacterised carminic acid biosynthetic pathway. The work detailed in this manuscript will be of interest to others and in particular should attract further research into the up-stream biosynthetic enzymes. In addition, the authors have laid the foundations for the production of carminic acid in an alternative host with their detailed and insightful work on the heterologous expression.

While the work in general is convincing, a couple of more points should be addressed to strengthen the conclusions.

Firstly, can it be demonstrated that the C-glycosyltransferase is indeed from an insect rather from a 'contaminating' endosymbiont bacterium? Presumably, some phylogenetic analysis will be sufficient to determine this point.

Secondly, the authors describe the final C-glycosylation step as a 'detoxification' step, however carminic acid is thought to act as defence compound. The authors should reconcile these two contradictory ideas in the discussion.

The level of detail in the paper should enable researchers to reproduce the work with comparative ease and in addition provides a great deal of technical information which should be of great help to an inexperienced enzyme biochemist. A reference or detailed method is required for the isolation of flavokermesic acid and dcll from *D.coccus* (pg 26, line 448).

This manuscript should be accepted after completion of the minor corrections detailed in this review.

Response to reviewers' comments:

Reviewer #1:

1) Figure 5: In (d) you show the MS and MS/MS data for the standards, but not the experiment. I would have liked to see the corresponding data from the CA and dcll peaks in (c) for comparison.

- As requested by the reviewer, we have replaced Figure 5 with a new figure which includes both MS and MS/MS data for the experimental samples: *K. vermilio* metabolite mix (panel A), pYES-DEST52 (panel B) and DcUGT2-Strep (panel C). We appreciate this comment.

2) 107/Supplementary table 1: reporting average coverage is not typically used to say which are the "highest expressed" transcripts. More typically these abundance values are normalized for gene length along the lines of RPKM (reads per kb sequence per million reads) or another related metric, since coverage values for a transcript can be biased by transcript length, splice variants, etc. While I don't think this will change the conclusions significantly, it would improve the presentation

- As suggested by the reviewer, we have now inserted an additional column in supplementary table 1 which depicts the RPKM values for the different *DcUGT* candidates identified. We agree that it improves the data presentation and have therefore revised following sentence in the manuscript to accommodate this change:

(p5)

"The four full-length sequences (*DcUGT1*, *DcUGT2*, *DcUGT4* and *DcUGT8*) were among the 21 highest expressed putative *UGT* transcripts in adult female *D. coccus* insects displaying an average coverage of 487, 820, 244 and 43, respectively (Supplementary Table 1)"

to

"The four full-length sequences (*DcUGT1*, *DcUGT2*, *DcUGT4* and *DcUGT8*) were among the 21 highest expressed putative *UGT* transcripts in adult female *D. coccus* insects displaying RPKM (Reads per Kilobase sequence per Million mapped reads) values of 108, 182, 54 and 10, respectively (Supplementary Table 1)."

3) 216: Is it more appropriate to say that the assembled DcUGT23 transcript lacked a stop codon, rather than the isolated transcript? I would think the lack of a stop codon more likely represents an assembly or sequencing artifact than a problem with the RNA transcript itself.

- We fully agree with reviewer #1 that it would be more accurate to state that it is the assembled transcript which lacks a stop codon and not the isolated transcript. We have therefore changed

(p10)

"The isolated transcript encoding *DcUGT23*, however, lacks a stop codon but the translated product of this partial transcript is 515 amino acids and thus assumed to be nearly full-length."

to

“The assembled transcript encoding *DcUGT23*, however, lacks a stop codon but the translated product of this partial transcript is 515 amino acids and thus assumed to be nearly full-length.”

4) 221: missing quote after X3-VL.

- We have clarified the referred paragraph and changed it:

(p10)

“The alignment of *DcUGT2* to several other membrane-bound UGTs shows that the enzyme contains a region between amino acids 46 and 56 which is 91 % identical to a conserved hydrophobic motif “LX2-RG-H-X3-VL described in *UGT2B7* from humans.”

to

“The alignment of *DcUGT2* to several other membrane-bound UGTs shows that the enzyme contains a region between amino acids 46 and 56 which corresponds to the conserved hydrophobic motif “LX2-RG-H-X3-VL” e.g. as described in human *UGT1A6*²³. The “LX2-RG-H-X3-VL” sequence region in *DcUGT2* is 91 % identical at the amino acid level to the corresponding sequence in *UGT2B7* from humans (Supplementary Fig. 3).”

We have added in the paragraph the correct citation:

23 Senay, C. *et al.* The importance of cysteine 126 in the human liver UDP-glucuronosyltransferase *UGT1A6*. *Biochim. Biophys. Acta, Protein Struct. Mol. Enzymol.* **1597**, 90-96 (2002).

This reference has also been added to the reference list.

5) 225: Histidine to asparagine is a relatively conservative mutation; it’s certainly fine to note the change at a likely catalytically important residue, but can you add a bit of context/interpretation to this statement? Is there anything known about the impact of this type of mutation at that site for function?

- We agree with the reviewer that the effect of this substitution would be of interest to address. Indeed, we did carry out protein modelling experiments of *DcUGT2* with an asparagine as well as a histidine residue at this position. No structural differences were found. A slight difference was observed using the I-Tasser server but not within this catalytic residue and also not enough to comment on. Since we cannot offer additional information to the readers on this matter, we have not modified the manuscript.

6) 251: Should be catalytically active

(p11)

- We have corrected “catalytic active” to “catalytically active”.

7) 372: Should be SMARTer

(p17)

- We have corrected “SMATer” to “SMARTer”.

8) Corrections and comments to the Method section.

I tried to examine the NCBI datasets, but they appear to not be publicly released yet. Thus, I was not able to independently check these data. I trust that the sequences will be released upon publication.

- Yes, reviewer#1 is indeed perfectly correct in this statement. The datasets have been deposited in NCBI. However, the sequences are currently not accessible but upon publication of the manuscript all datasets will be released immediately and become automatically available to the public.

The transcriptome sequencing and assembly methods need more detail. For reproducibility purposes, please provide specifics on the quality filtering/trimming settings as well as assembly algorithm settings beyond just ‘defaults’ – such defaults are not particularly obvious to others who do not have the commercial package you used. Also, I am not familiar with the IOGMA software; I presume you used a prokaryotic HMM model because the package would otherwise look for splicing sites in the assembled transcriptome? Or is there another reason why a eukaryotic model was not used?

- We realize that the method description regarding the transcriptome assembly provided in the original version of the manuscript was inadequate and have now added the following requested information in the method section of the manuscript to clarify our approach:

(p15)

“Quality-based read trimming was performed based on Phred scores, using a modified Mott-trimming algorithm with a limit of 0.05 and a maximum of 2 ambiguous bases· reads⁻¹ after trimming. More details about the Mott-trimming algorithm used by CLC-bio can be found in online documentation³⁷. *De novo* transcriptome assembly was carried out using the de Bruijn graph algorithm in the CLC bio Genomic Workbench. Settings were a word size of 20, a bubble size of 50 and a minimum contig length of 200. After assembly, the reads were mapped back to the contigs with the following mapping parameters: mismatch cost=2, insertion cost=3, deletion cost=3, length fraction = 0.5 and similarity fraction = 0.8.”

Accordingly, we have updated our reference list with:

37 CLC bio, CLC Genomics Workbench Manual - Quality trimming,
http://resources.qiagenbioinformatics.com/manuals/clcgenomicsworkbench/551/index.php?manual=Quality_trimming.html (2012).

- We used a bacterial gene finder tool based on a Hidden Markov Matrix (HMM) for the identification of the transcribed genes in the assembled RNA contig sequences. In the current study, we only analyzed the *D. coccus* polyadenylated RNA, which would imply that splicing events would have already occurred. Therefore, it was more simple to use a prokaryotic gene finder approach than the available eukaryotic gene finder method which would be suitable for assembly of eukaryotic genomes, including the genome from *D. coccus*.

We agree with reviewer#1 that the rationale behind choosing a prokaryotic gene finder tool over a eukaryotic gene finder approach was not evident. Therefore, following paragraph has been extended in the manuscript:

(p15)

“Putative genes were identified using the hidden Markov-Matrix-based prokaryote gene-finder tool in IOGMA v. 10 (Genostar, Grenoble, France).”

to

“Putative genes were identified using the Hidden Markov Matrix-based prokaryote gene finder tool in IOGMA v. 10 (Genostar, Grenoble, France). This approach was regarded to be more simple than using the eukaryote gene finder tool since only polyadenylated RNA, in which splicing events are presumed to already have occurred, was analyzed.”

Also, can you specify which relevant Pfam families you were looking for when looking through the assembled contigs?

- As requested we have now specified the relevant Pfam family we used in our comparison to identify putative *UGT* candidates among the assembled contigs. Accordingly following sentence have been altered:

(p15-16)

“Annotation of putative UDP-glucosyltransferase (UGT) genes was carried using both the nucleotide and translated protein sequences in a BLAST comparison with the Genbank sequence database (National Center for Biotechnology Information, NCBI) and by similarity comparison to Pfam databases of protein families³⁶.”

to

“Annotation of putative UDP-glucosyltransferase (UGT) genes was carried using both the nucleotide and translated protein sequences in a BLAST comparison with the Genbank sequence database (National Center for Biotechnology Information, NCBI) and by similarity comparison to the UDPGT (UDP-glucuronosyl and UDP-glucosyl transferase) Pfam protein family (PF00201)³⁸”

While you did confirm the full-length sequence by RACE, was there any hint from the RACE clones, or from re-aligning the transcriptome reads to the transcriptome assemblies, that there might exist other isoforms of DcUGT2?

- We did not observe any isoforms nor splice variants of *DcUGT2* after sequencing a number of RACE clones. A re-alignment of the reads to the assembled transcriptome also did not support the existence of other DcUGT2 isoforms in *D. coccus*. Additionally, we have now sequenced and assembled the *D. coccus* genome (which will be published in another manuscript) and this also does not indicate that *D. coccus* would have any additional *DcUGT2* variants. We cannot completely exclude the existence of DcUGT2 isoforms arising from post-translational modifications of the protein. However, western blot analysis of crude *D. coccus* protein (soluble and membrane-bound proteins, Figure 8) only afforded a single immunoreactive band with our peptide-specific DcUGT2 antibody.

Reviewer #2:

1) Figure 1. Malonyl-CoA is incorrectly drawn as Acetoacetyl-CoA.

- We have replaced Figure 1 with a new figure in which the acetoacetyl-CoA structure has been corrected to malonyl-CoA. We are really grateful that the reviewer discovered this unfortunate mistake.

2) Page 6. “The four full-length DcUGT cDNAs were expressed in *S. cerevisiae* and *Aspergillus nidulans* without codon optimization. No functional UGT activity was obtained as monitored by the absence of changes in the LC-MS and TLC profiles following incubation of soluble and microsomal protein extracts from the transformed heterologous hosts with UDP-glucose or [14C]-UDP-glucose, respectively, and the putative substrates flavokermesic acid and kermesic acid when compared to incubation with the un-transformed host controls (data not shown).” Why is there a discrepancy between these unsuccessful heterologous expression experiments and later successful ones? Is it the codon optimization? This disagreement is not explicitly discussed.

- We do think that codon optimization was vital for obtaining successful expression of the tested *DcUGT* cDNAs. Initially, the four native *DcUGT* cDNA sequences were expressed without and with a C-terminal Strep-tag II in *S. cerevisiae* and *A. nidulans*. Transformants were confirmed by both PCR and DNA sequencing of the product. But when we tested for heterologous protein production using total extracted protein from the cultured transformants (after induced expression) in western blot analysis with an anti-StrepII antibody, no immunoreactive proteins could be detected. Although unable to observe any immunoreactive proteins, we still proceeded to measure whether there were any new UGT activity produced upon induced expression. The reason for this was that we initially thought that the heterologous DcUGT proteins could have been produced at levels that were below our chemiluminescence detection limit. In addition, non-epitope tagged DcUGT versions were also tested in case the tag had any negative effects on the activity of the enzymes. As

both non-tagged and epitope-tagged versions of the native *DcUGT* cDNAs did not produce any novel UGT activity upon induced expression. We are therefore convinced, that non-optimal codon usage of the native cDNA sequences is the reason no protein was produced and no activity obtained. The lack of enzyme activity might be a result of an either hampered transcription of these native *DcUGT* cDNA sequences or an instability/degradation of the foreign *DcUGT* transcripts in the heterologous host which also would lead to a lack of heterologous *DcUGT* protein production. We did not do RT-PCR to further investigate whether this was the case since codon optimization appeared to solve our issue.

To address the discrepancy between the unsuccessful and successful heterologous expression we have changed the following paragraph in the manuscript:

(p5-6)

“The four full-length *DcUGT* cDNAs were expressed in *S. cerevisiae* and *Aspergillus nidulans* without codon optimization. No functional UGT activity was obtained as monitored by the absence of changes in the LC-MS and TLC profiles following incubation of soluble and microsomal protein extracts from the transformed heterologous hosts with UDP-glucose or [¹⁴C]-UDP-glucose, respectively, and the putative substrates flavokermesic acid and kermesic acid when compared to incubation with the un-transformed host controls (data not shown). Flavokermesic acid and kermesic acid were supplied in the form of an isolated metabolite fraction from *Kermes vermilio*, a scale insect species incapable of producing dcll and carminic acid.”

to

“An attempt to express the four full-length native *DcUGT* cDNAs were carried out in *S. cerevisiae* and *Aspergillus nidulans* with and without a C-terminal Strep-tag II (Strep) epitope. Transformants were confirmed by PCR followed by DNA sequencing of the amplified product but no functional UGT activity could be measured. The UGT activity was monitored by the absence of changes in the LC-MS and TLC profiles following incubation of soluble and microsomal protein extracts from the transformed heterologous hosts in the presence of UDP-glucose or [¹⁴C]-UDP-glucose, respectively, and the putative substrates flavokermesic acid and kermesic acid when compared to incubation with the un-transformed host controls. Flavokermesic acid and kermesic acid were supplied in the form of an isolated metabolite fraction from *Kermes vermilio*, a scale insect species incapable of producing dcll and carminic acid. The lack of a novel UGT activity prompted us to test for heterologous protein production after induced expression of the epitope-tagged *DcUGT* versions. Western blot analysis of total proteins extracted from cultured transformants did not uncover any immunoreactive proteins. Thus, the absence of heterologous UGT activity was ascribed to either non-optimal codon usage of the native *DcUGT* cDNA sequences, hampered transcription or an instability/degradation of the foreign *DcUGT* transcripts.”

- 3) Page 7. “A protein fraction eluted with 100 mM NaCl was separated by SDS-PAGE and the proteins migrating in the 50 to 75 kDa region were subjected to in gel trypsin digestion, LC-MS/MS- based amino acid sequencing of the fragments obtained and database searching (Fig.

2c).” Why was this fraction selected? Fractions eluting at higher NaCl concentrations also have flavokermesic acid and kermesic acid-specific glycosylation activity

- Fraction #1 was selected basically because it contained the desired activity and when analyzed by SDS-PAGE followed by Coomassie staining, there were fewer proteins present in this fraction in the size region spanning 50 to 75 kDa compared to the other fractions (Supplementary 1a). Thus, we hoped that by *in gel* digesting this size region in fraction #1 and subsequently analyzing it by LC-MS/MS, we would reduce the number of non-relevant proteins and increase our chances in identifying the UGT, responsible for the observed activity.

To make this point clear in the manuscript we have changed the following paragraph:

(p6-7)

“A protein fraction eluted with 100 mM NaCl was separated by SDS-PAGE and the proteins migrating in the 50 to 75 kDa region were subjected to *in gel* trypsin digestion, LC-MS/MS-based amino acid sequencing of the fragments obtained and database searching (Fig. 2c). This mass region was chosen as UGT enzymes typically are considered to be within this mass range.”

to

“UGT enzymes have masses within the range of 50 to 75 kDa. Based on the presence of the desired enzyme activity and SDS-PAGE analysis, protein fraction 1 was selected for further analysis. In comparison to other active protein fractions, it contained fewer proteins in the 50 to 75 kDa mass region (Supplementary Fig. 1). We expected that a reduced number of non-relevant proteins would optimize identification of the UGT responsible for the observed activity. Thus, protein fraction 1 was separated by SDS-PAGE and the proteins migrating in the 50 to 75 kDa region were subjected to *in gel* trypsin digestion, LC-MS/MS-based amino acid sequencing of the fragments obtained and database searching (Fig. 2c).”

4) Figure 3. To guide the reader, the texts “DcUGT2-Strep”, “DcUGT4-Strep” and “DcUGT5-Strep” should be added above their matching gel/blot pair.

- We have replaced Figure 3 with a new figure which includes the protein name over their matching gel/blot pair and agree this conveys the results in a more clear way.

5) Page 10. “As observed from the empty vector control, *S. cerevisiae* also possesses endogenous glycosylation activities, yielding a [14C]-product with a similar Rf value as carminic acid in the applied TLC system (Fig. 4).” Why are the DcUGT5-Strep lanes’ unknown [14C]-product bands elongated like the DcUGT2-Strep lanes’ bands and unlike the pYES-DEST52 lanes’ bands?

- We agree that it is interesting that the unknown [14C]-product band observed in the pYES-DEST52 lane is much more compact compared to the corresponding band observed in the DcUGT2-Strep and DcUGT5-Strep lanes. One might conclude that this is the result of several different [14C]-labelled products co-migrating together with similar Rf values which then

would imply that both DcUGT2-Strep and DcUGT5-Strep are able to make such unknown [14C]-products in addition to the endogenous glucosylation activity observed in yeast. However, we do not think this is likely. We think that the observed band size difference of the unknown [14C]-product is caused by the “empty” pYES-DEST52 vector. The commercial pYES-DEST52 vector is a gateway destination plasmid and contains a *ccdB* gene. This gene will be expressed in the yeast in the negative control strain, and although there is no literature which reports that the *ccdB*-encoded protein would induce adverse effects or be toxic for yeast, we think the *ccdB* protein might be partially inhibiting the endogenous glucosylation activity in an irreversible manner. At present, the *ccdB* protein is only characterized to be toxic for bacteria because it interacts with bacterial gyrase, thus preventing bacterial growth. To make it clear in the manuscript we have removed the word “empty” in front of all negative pYES-DEST52 controls to avoid the impression that the yeast strain containing this plasmid is completely devoid of any galactose-induced heterologous protein production. In reflection one might argue whether it would have been better to use the non-transformed yeast as a negative control. However, in such case it would be impossible to know whether the effects we observed were caused by the gene we expressed or some random effect of the plasmid. The inclusion of an additional non-transformed yeast control would not change the conclusion of our manuscript.

6) Figure 5. As in Figure 3, to guide the reader, the texts “pYES-DEST52” and “DcUGT2-Strep” should be added above their matching chromatograms.

- We have replaced Figure 5 with a new figure which includes “pYES-DEST52” and “DcUGT2-Strep” above their matching chromatograms and agree this conveys the results in a more clear way.

7) Page 12. “To characterize the kinetic properties of DcUGT2, yeast microsomal membranes containing heterologously-produced DcUGT2 were solubilized using reduced Triton X-100 and DcUGT2 was affinity-purified by its Strep-tag II (Fig. 6). In vitro tests of the isolated Strep-tagged DcUGT2 showed that its enzymatic activity was lost upon isolation (Fig. 6).” The glycosylated form of DcUGT2 looks like it is present in the flow-through and not in the eluate. Were in vitro tests performed using the flow-through?

- We do not agree that the DcUGT2-Strep only was present in the flow-through and not the eluate. In figure 6, only one protein band is observed with Coomassie staining and immunodetection in the eluate and this corresponds well to the size of DcUGT2-Strep. We think that the reviewer might have had some difficulties seeing this band in the figure because of the low content of protein. We have therefore replaced the entire figure with a new figure, where the overall contrast has been increased in the same way for all lanes. In this way the presence of a protein band in the eluate is apparent.
- We did perform *in vitro* activity test on the flow-through and indeed did observe some enzyme activity as was also the case with the solubilized microsome (SM) sample. We assume that the loss of enzyme activity could have occurred during the binding of DcUGT2-Strep to the Strep-tag affinity matrix. It may simply also reflect that upon microsomal solubilizations, some DcUGT2-Strep protein molecules are retained in an active conformation where the Strep-tag is not exposed and therefore cannot bind to the affinity matrix while other DcUGT2-Strep molecules get inactivated because of conformational changes that expose the Strep-tag. As a consequence, only the inactivated DcUGT2-Strep protein molecules would then bind and elute from the column during purification.

8) **Page 18. “In vitro activity assays using yeast microsomes containing the full-length DcUGT2-Strep, truncated DcUGT2-Strep versions or microsomes from yeast harboring the empty vector showed that only DcUGT2 versions expressed with the N-terminal signal peptide were catalytic active when compared to the empty vector control (Fig. 10). Although functionally active, the truncated DcUGT2ΔMD-Strep was not as efficient as DcUGT2-Strep.” The production of carminic acid was reduced by two orders of magnitude and production of dclI was abolished. This paragraph does not accurately describe how impaired DcUGT2ΔMD-Strep is relative to DcUGT2-Strep.**

- We agree with the reviewer that the paragraph in the original manuscript did not accurately describe how impaired DcUGT2ΔMD-Strep was relative to DcUGT2-Strep. We have therefore as requested modified the following paragraph:

(p11)

“In vitro activity assays using yeast microsomes containing the full-length DcUGT2-Strep, truncated DcUGT2-Strep versions or microsomes from yeast harboring the pYES-DEST52 vector showed that only DcUGT2 versions expressed with the N-terminal signal peptide were catalytically active when compared to the vector control (Fig. 10). Although functionally active, the truncated DcUGT2ΔMD-Strep was not as efficient as DcUGT2-Strep. This was demonstrated by a reduced relative production of carminic acid and no detectable production of dclI when microsomes containing DcUGT2ΔMD-Strep were incubated in the presence of UDP-glucose and kermesic acid and flavokermesic acid as the respective substrates (Fig. 10)”

to

“In vitro activity assays using yeast microsomes containing the full-length DcUGT2-Strep, truncated DcUGT2-Strep versions or microsomes from yeast harboring the pYES-DEST52 vector showed that only DcUGT2 versions expressed with the N-terminal signal peptide were catalytically active when compared to the vector control (Fig. 10). Although functionally active, the truncated DcUGT2ΔMD-Strep was not as efficient as DcUGT2-Strep. The production of carminic acid was reduced by two orders of magnitude and production of dclI was abolished when compared with DcUGT2-Strep (Fig. 10)”

9) **Figure 10. As in Figures 3 and 5, to guide the reader, the texts “pYES-DEST52”, “DcUGT2-Strep”, “DcUGT2ΔMD-Strep” and “ΔSP-DcUGT2ΔMD-Strep” should be added above their matching chromatograms.**

- We have replaced Figure 10 with a new figure which includes “pYES-DEST52”, “DcUGT2-Strep”, “DcUGT2ΔMD-Strep” and “ΔSP-DcUGT2ΔMD-Strep” above their matching chromatograms and agree this conveys the results in a more clear way.

10) **Page 20. “The native *D. coccus* DcUGT2 was shown to be subject to heavy glycosylation. Whether such post-translational modifications are required for its catalytic activity is**

uncertain.” Could the three putative asparagines for N-glycosylation be mutated to glutamines to test whether this modification is required for catalytic activity?

- We agree that it could be interesting to see whether the catalytic activity would be affected by mutating the asparagines in the three putative N-glycosylation sites of DcUGT2 to glutamine. Although, site directed mutagenesis of asparagine to glutamine is a widely used approach to study N-linked protein glycosylation, it may not necessary give us more information. Glutamine is chemically very similar to that of asparagine, but it is still slightly larger as it contains an additional single methylene group in the side chain compared to asparagine. Thus, we cannot completely exclude that by performing such conservative amino acid substitutions, it would not change the overall conformation of the DcUGT2 protein. As a consequence, it would be impossible for us to then discern whether a negative activity result obtained with such an altered enzyme is caused by a simple change in protein conformation rather than lack of N-glycosylation.

As mentioned in the manuscript (discussion, last paragraph), the goal for constructing the different truncated forms of DcUGT2 was to see whether it would be possible to generate a catalytic active DcUGT2 which would be suitable for heterologous production of carminic acid in prokaryotic host organisms. We demonstrate in the current study that DcUGT2 needs the ER-targeting signal to be active. We therefore reason that it may not only be post-translational glycosylation but also other ER conditions which are crucial for the activity of this protein. To further identify whether post-translational glycosylation is required for catalytic activity seems to be futile in this context as this will still not solve our issue of generating an active soluble DcUGT2 version which is not restricted to the ER.

Reviewer #3:

1) While the work in general is convincing, a couple of more points should be addressed to strengthen the conclusions.

Firstly, can it be demonstrated that the C-glucosyltransferase is indeed from an insect rather from a ‘contaminating’ endosymbiont bacterium? Presumably, some phylogenetic analysis will be sufficient to determine this point.

Secondly, the authors describe the final C-glycosylation step as a ‘detoxification’ step, however carminic acid is thought to act as defence compound. The authors should reconcile these two contradictory ideas in the discussion.

- The origin of the C-glucosyltransferase DcUGT2 is indeed a valid question. There are several lines of evidence that the DcUGT2 is eukaryotic-derived and comes from *D. coccus* and not a prokaryote. Firstly, the DcUGT2 nucleotide sequence was identified from a transcriptome that was assembled using sequenced reads generated from polyadenylated RNA isolated from *D. coccus*. As we used an oligo-dT primer (that targets eukaryotic poly (A) tails) and a reverse transcriptase (RT² Easy First Strand Kit, Qiagen) to convert the polyadenylated RNAs into cDNA, prior to sequencing, it should exclude the possibility of RNA contamination from any endosymbiotic bacteria. Secondly, we performed RACE PCR to subsequently confirm the DcUGT2 cDNA sequence using polyadenylated RNA isolated from *D. coccus*. We employed the SMARTer RACE 5’/3’ kit (Clontech) where first-strand cDNA synthesis is primed using a modified oligo (dT) primer. This means that any RACE products, amplified from this template, would again be derived from a eukaryotic-derived polyadenylated RNA. Thirdly, we have now sequenced and assembled the *D. coccus* genome (which will be published in another manuscript). The DcUGT2 gene is indeed present in the *D. coccus* genome. In the

assembled genome, the DcUGT2 gene contains 3 introns and although some bacterial introns are known, they are not found in most bacterial genes. Thus, we trust that the results, provided in the manuscripts, are convincing enough to unequivocally argue that DcUGT2 is encoded by *D. coccus* and not a bacterial endosymbiont.

- The reviewer has a point, that the idea of carminic acid acting as defence compound might seem contradictory to the idea of the C-glycosylation step serving as a detoxification mechanism. To address this we inserted the following paragraph in the discussion:

(p12-13)

“Generally, glycosylation serves to stabilize labile aglycons, to increase their solubility, facilitate compartmentalized storage and to reduce their bioactivity/autotoxicity. This is the reason e.g. many plant defense compounds are stored as glucosides. In some cases, the sugar moiety is cleaved off to activate and jack-up the efficacy of the defense system upon demand. In *D. coccus*, the C-glycosylation step would be expected to facilitate transport, packing and safe storage of carminic acid. Carminic acid is envisioned to serve as a defense compound due to its feeding-deterrent properties towards ants³. Storage of toxic constituents is a challenge that not only *D. coccus* but all organisms need to handle and master if they want to use them as part of their defense systems towards predators and pests.”

2) The level of detail in the paper should enable researchers to reproduce the work with comparative ease and in addition provides a great deal of technical information which should be of great help to an inexperienced enzyme biochemist. A reference or detailed method is required for the isolation of flavokermesic acid and dcll from *D.coccus* (pg 26, line 448).

- We agree with reviewer#3 that the technical information presented in the original manuscript was not very detailed with respect to the isolation of flavokermesic acid and dcll. We have now expanded the text to read as follows:

(p20)

“Flavokermesic acid and dcll were isolated by extracting dried and ground *D. coccus* with methanol-water (1:1 (v/v)) adjusted to pH 3 with formic acid. The extract was partitioned three times between ethyl acetate and the ethyl acetate phases were collected, combined and concentrated *in vacuo*. The extract was then subjected to ion-exchange chromatography using a column packed with Septra NH₂ functionalized silica (Phenomenex). The column was equilibrated in acetonitrile-water (1:1 (v/v)) containing 10 mM ammonium formate prior to application of the extract. Subsequently, the column was washed with the equilibration solvent followed by elution of flavokermesic acid and dcll with acetonitrile-water (1:1 (v/v)) adjusted to pH 11 with ammonium hydroxide. Final isolation was achieved on a column packed with Isolute diol functionalized silica (Biotage), using a step-wise elution gradient from dichloromethane-to-ethyl acetate-to-methanol, to afford flavokermesic acid and dcll. Their molecular structures were verified by 1D and 2D NMR.”

REVIEWERS' COMMENTS:

Reviewer #1 (Remarks to the Author):

The authors have responded reasonably to my concerns and those of the other reviewers and made the appropriate corrections. At this point I find no other major fault in the study.

Reviewer #3 (Remarks to the Author):

I am satisfied with the revision of the manuscript.